# FedProTIP: Task-Agnostic Federated Continual Learning via Replay-Free Gradient Projection

**Seohyeon Cha**[*]                                          *seohyeon.cha@utexas.edu*
*Department of Electrical and Computer Engineering*
*The University of Texas at Austin*

**Huancheng Chen**[*]                                          *huanchengch@gmail.com*
*Department of Electrical and Computer Engineering*
*The University of Texas at Austin*

**Haris Vikalo**                                          *hvikalo@ece.utexas.edu*
*Department of Electrical and Computer Engineering*
*The University of Texas at Austin*

**Reviewed on OpenReview:** *https://openreview.net/forum?id=GW4aw0fUKC*

## Abstract

Federated continual learning (FCL) enables collaborative model training across distributed clients on sequentially arriving tasks without revisiting past data. However, existing approaches often suffer from catastrophic forgetting, rely on replay buffers or generative models that may violate privacy constraints, or assume knowledge of task identities during inference. We propose FedProTIP (Federated Projection-based Continual Learning with Task Identity Prediction), a replay-free FCL framework that maintains shared task-specific feature subspaces across clients. Each client extracts low-rank core bases from intermediate activations using randomized singular value decomposition, capturing dominant feature directions associated with the current task. These bases are transmitted to the server and aggregated to construct global task subspaces that capture shared feature directions across clients without requiring data sharing. During training, client updates are projected onto the orthogonal complement of previously learned subspaces to reduce cross-task interference and mitigate catastrophic forgetting. The learned subspaces are also reused during inference to estimate task identity via subspace relevance, enabling task-agnostic prediction without requiring explicit task labels. Experiments on CIFAR100, ImageNet-R, and DomainNet demonstrate that FedProTIP consistently outperforms state-of-the-art federated continual learning baselines while maintaining lower training time, memory footprint, and communication cost. Our code is available at GitHub.

## 1 Introduction

Federated learning (FL) (McMahan et al., 2017), where client devices collaboratively train a global model without sharing private data, has emerged as an alternative to centralized learning. Most FL systems assume static local datasets and a single inference task per client. In practice, however, devices (e.g., phones, smart glasses) collect data for multiple evolving tasks and must continually adapt their models over time. Limited storage often forces clients to discard data from earlier tasks, resulting in a continual learning (CL) setting in which models must incorporate new information without direct access to prior data. This setting exacerbates *catastrophic forgetting* (McCloskey & Cohen, 1989), i.e., the degradation of performance on previously learned tasks. In federated environments, forgetting is further amplified by statistical heterogeneity across clients, which induces drift between local updates and the global objective.

---

[*]Equal Contribution.

A growing body of work seeks to adapt conventional continual learning strategies to federated settings. Existing approaches typically fall into three categories: (1) *replay-based* methods (Dong et al., 2022; Liu et al., 2023; Dai et al., 2024; Li et al., 2024c;a), which retain past data; (2) *generation-based* methods (Qi et al., 2023; Zhang et al., 2023; Tran et al., 2024; Liang et al., 2024; Yu et al., 2024), which train generative models to synthesize past examples; and (3) *regularization-based* methods (Yoon et al., 2021; Ma et al., 2022; Li et al., 2024b; Lee et al., 2024), which constrain parameter updates to preserve prior knowledge. In federated deployments, each class of methods faces practical limitations. Replay-based schemes require storing historical data, raising privacy and storage concerns. Generation-based methods rely on server-side generative models, increasing communication and computational overhead. Regularization-based approaches often introduce additional local training complexity and may struggle under severe client heterogeneity.

Recently, gradient projection methods such as GPM (Saha et al., 2021) have shown effectiveness at mitigating forgetting in centralized continual learning by projecting gradients onto subspaces orthogonal to representations of prior tasks. However, GPM requires centralized access to activation statistics, making it incompatible with federated data constraints. FOT (Bakman et al., 2024) extends the GPM idea to FL by having clients transmit randomized activation sketches that are combined through secure aggregation, after which the server extracts projection subspaces. However, enforcing orthogonality only after aggregation does not constrain local optimization trajectories. During local training, clients update parameters without projection, and gradients may move the model into directions that interfere with prior tasks. This effect is amplified under heterogeneous client data, where interference can accumulate before aggregation. Projecting only the final aggregated update therefore cannot fully eliminate cross-task interference.

In this work, we propose **FedProTIP (Federated Projection-based Continual Learning with Task Identity Prediction)**, a federated continual learning framework that enforces projection constraints during local optimization and supports task-agnostic inference. Rather than projecting only the aggregated global update at the server, FedProTIP applies projected gradient descent during local client optimization using a globally shared subspace that captures directions associated with previously learned tasks. After completing a task, clients extract compact low-rank bases from their learned representations and transmit only these bases to the server, which aggregates them into a global orthonormal subspace and broadcasts it for subsequent training rounds. The proposed design mitigates cross-task interference at its source while communicating only compact low-rank subspace bases. We analyze the projected local training dynamics and derive task-wise convergence and forgetting bounds that reveal how geometric properties of the learned subspaces influence the stability–plasticity trade-off across tasks. In addition, FedProTIP removes the assumption that task identities are available at inference. Specifically, we introduce a task identity prediction (TIP) mechanism that leverages learned subspaces to estimate task relevance for each test input and route predictions accordingly. This enables effective task-agnostic federated continual learning without replay buffers, generative models, or auxiliary classifiers. Extensive experiments across multiple benchmarks demonstrate consistent improvements over existing FCL methods under both heterogeneous and task-agnostic settings. In particular, we show that enforcing projection only after aggregation degrades sharply under strong client heterogeneity, whereas local projected descent remains stable.

The main contributions of this paper are as follows:

- We propose *FedProTIP*, a federated continual learning framework that enforces subspace-based gradient projection during local client optimization. By constraining local updates rather than projecting only the aggregated global update, FedProTIP mitigates cross-task interference under client heterogeneity. To remain communication-efficient, it transmits only compact low-rank core bases to construct the global projection subspace, avoiding raw activation sharing.

- We introduce a *task identity prediction (TIP)* mechanism based on subspace relevance alignment. TIP infers task identity at inference time without replay buffers, generative models, or auxiliary classifiers, enabling task-agnostic federated continual learning.

- We provide extensive empirical evaluation on CIFAR-100, ImageNet-R, and DomainNet under heterogeneous data partitions. FedProTIP improves average accuracy by 4.3%–47% over prior FCL methods while maintaining low forgetting and reduced communication and memory overhead.

## 2 Related Work

### 2.1 Federated Continual Learning

Federated continual learning (FCL) addresses the problem of learning a sequence of tasks on data decentralized across clients. An early FCL approach, FedWeIT (Yoon et al., 2021), decomposes model parameters into task-generic and task-specific components, focusing on a task-incremental setting where the task IDs are known during inference. CFeD (Ma et al., 2022) relies on knowledge distillation using a surrogate dataset shared between the server and clients. GLFC (Dong et al., 2022; 2023) mitigates catastrophic forgetting by combining class-aware gradient compensation with class-semantic relation distillation, but relies on storing examples from previous tasks. Subsequent works (Liu et al., 2023; Dai et al., 2024; Li et al., 2024c;a) reduce replay memory requirements and, in some cases, provide convergence analysis (Keshri et al., 2025).

Recently, several FCL methods have leveraged generative models to replace stored examples with synthetic data. FedCIL (Qi et al., 2023) employs a GAN with an auxiliary classifier to enable generative replay, mitigating forgetting while aggregating global knowledge across clients. TARGET (Zhang et al., 2023) and MFCL (Babakniya et al., 2024) introduce data-free knowledge distillation using synthetic examples to transfer knowledge from a previously trained global model to client models. LANDER (Tran et al., 2024) further incorporates label text embeddings from pretrained language models as anchors to improve the semantic quality of generated samples and enhance resistance to forgetting. Although effective, generative approaches introduce additional computational overhead as image resolution increases, and may raise privacy concerns (Liu et al., 2024). In general, existing FCL methods face practical challenges in real-world deployments due to privacy and resource constraints. Many assume that task identity is available at inference, store exemplars from previous tasks, or rely on generative replay to synthesize past data. In contrast, FedProTIP is designed for task-agnostic inference, i.e., the settings where task labels are unavailable at test time, and operates without replay buffers, generative models, or auxiliary task classifiers. Instead, it leverages lightweight subspace representations for both knowledge retention and task-identity prediction. This formulation connects to the broader literature on class-incremental learning (CIL). (Kim et al., 2022b) shows that strong CIL performance requires both within-task classification and accurate task-identity prediction. While centralized approaches address task-agnostic inference through out-of-distribution detection (Kim et al., 2022b;a), per-class classifiers or generative models (Zając et al., 2024), or supervised contrastive learning with nearest-class-mean classifiers (Mai et al., 2021), these strategies do not readily extend to federated settings. In contrast, FedProTIP integrates task-identity prediction directly into a replay-free federated continual learning framework.

### 2.2 Gradient Projection in Continual Learning

Gradient projection methods (Zeng et al., 2019; Farajtabar et al., 2020; Chaudhry et al., 2020) for continual learning mitigate forgetting by updating model parameters in directions orthogonal to those associated with previous tasks, thereby eliminating the need to store raw data or train generative models. GPM (Saha et al., 2021) extends this line of work by extracting low-dimensional subspaces from prior-task representations and constraining subsequent updates to be orthogonal to the corresponding subspaces. Follow-up works such as TRGP (Lin et al., 2022b), CUBER (Lin et al., 2022a), SGP (Saha & Roy, 2023), DualGPM (Liang & Li, 2023a) and DualLoRA (Chen et al., 2024) relax the strict orthogonality requirement to trade off stability and plasticity. On a related note, parameter-efficient continual learning for pretrained models commonly relies on low-rank adapters whose task-specific updates are isolated (e.g., through subspace or orthogonality constraints) to mitigate cross-task interference (Liang & Li, 2024; Chen et al., 2024). However, translating gradient-projection ideas to federated continual learning is challenging because task information is distributed across clients and communication is limited. Recent attempts take two distinct directions. TAPGP (Ke et al., 2025) adopts a parameter-efficient prompt-tuning approach and mitigates interference by projecting prompt gradients to be orthogonal to subspaces induced by prior-task virtual data and prompts. This avoids transmitting raw embeddings, but relies on a virtual replay pipeline that may impose nontrivial computational overhead and introduce additional privacy risks. In contrast, FOT (Bakman et al., 2024) constructs projection subspaces at the server from randomized activation sketches combined through secure aggregation, and applies projection only after client updates have been aggregated. While FOT provides formal privacy guarantees, it incurs substantial communication overhead and does not constrain local optimization trajectories. Moreover,

FOT is evaluated in settings where the task identity is available at inference, a strong assumption in many practical deployments. FedProTIP addresses these limitations by enforcing projection directly during local client optimization while communicating compact low-rank subspace information. It avoids sharing raw feature embeddings and does not rely on virtual replay or auxiliary generators. As a result, it supports task-agnostic inference without requiring task IDs at test time, while remaining communication- and computation-efficient.

## 3  Background and Problem Setup

### 3.1  Problem Formulation

We consider the problem of training a global model sequentially on streaming data $\mathcal{D}^{(t)} = \{\mathbf{x}_i^{(t)}, \mathbf{y}_i^{(t)}\}_{i=1}^{|\mathcal{D}^{(t)}|}$ distributed across $K$ client devices such that $\mathcal{D}^{(t)} = \mathcal{D}_1^{(t)} \cup \cdots \cup \mathcal{D}_K^{(t)}$. In the *domain-incremental* setting, the input distributions of two tasks, $\mathcal{X}^{(t_1)}$ and $\mathcal{X}^{(t_2)}$, are significantly different, while the label space may remain the same. In the *class-incremental* setting, the label sets of any two tasks are disjoint, i.e., $\mathcal{Y}^{(t_1)} \cap \mathcal{Y}^{(t_2)} = \emptyset$ for all $t_1 \neq t_2$. When learning a new task, data from earlier tasks is assumed to be inaccessible. The goal of federated continual learning is to obtain a global model $\mathbf{W}^{(T)}$ that minimizes the average empirical loss across $T$ tasks,

$$\min_{\mathbf{W}} \frac{1}{T} \sum_{t=1}^{T} \sum_{k=1}^{K} p_k^{(t)} \mathcal{L}_t(\mathbf{W}, \mathcal{D}_k^{(t)}), \tag{1}$$

where $p_k^{(t)}$ denotes the weight assigned to client $k$ on task $t$ (e.g., proportional to local data size), and $\mathcal{L}_t$ is the empirical loss for task $t$ on local data. During inference, **task identities are not revealed** to the model.

### 3.2  Gradient Projection Memory (GPM)

Gradient projection memory (Saha et al., 2021) is a replay-free CL scheme that requires storing only a set of core bases $\boldsymbol{\Phi}_l^{(1:t)}$ extracted from layer-wise activations after fine-tuning the model on $t$ tasks. Specifically, let $\mathbf{W}_l^{(t)}$ denote the parameters of layer $l$ within $\mathbf{W}^{(t)}$ after training on task $t$, and let $\mathbf{a}_l^{(t)} \in \mathbb{R}^{d_l \times m}$ represent the input activations to layer $l$ for $m$ training samples $\mathbf{x}^{(t)}$, where $d_l$ is the dimensionality of the activations. By applying singular value decomposition (SVD), GPM extracts a set of orthonormal bases $\boldsymbol{\Phi}_l^{(t)} \in \mathbb{R}^{d_l \times r_l^{(t)}}$ that span the dominant subspace of task $t$ activations and aggregates them with the existing bases $\boldsymbol{\Phi}_l^{(1:t-1)}$. During training on the $(t+1)$-th task, the parameter update for layer $l$, denoted $\Delta \mathbf{W}_l^{(t+1)}$, is projected onto the orthogonal complement of the subspace spanned by $\boldsymbol{\Phi}_l^{(1:t)}$,

$$\Delta \tilde{\mathbf{W}}_l^{(t+1)} \leftarrow \mathbf{Proj}_{\perp \boldsymbol{\Phi}_l^{(1:t)}} \left( \Delta \mathbf{W}_l^{(t+1)} \right). \tag{2}$$

Let $\mathbf{h}_l^{(\tau)} = \boldsymbol{\sigma}_l(\mathbf{W}_l^{(T)} \cdot \mathbf{a}_l^{(\tau)})$ denote the output activations for task $\tau$ ($\tau < T$) after training on $T$ tasks, where $\boldsymbol{\sigma}_l(\cdot)$ is the activation function at layer $l$. It follows from Eq. 2 that

$$\mathbf{h}_l^{(\tau)} = \boldsymbol{\sigma}_l \left( \mathbf{W}_l^{(\tau)} \cdot \mathbf{a}_l^{(\tau)} + \sum_{t=\tau+1}^{T} \Delta \tilde{\mathbf{W}}_l^{(t)} \cdot \mathbf{a}_l^{(\tau)} \right) \approx \boldsymbol{\sigma}_l \left( \mathbf{W}_l^{(\tau)} \cdot \mathbf{a}_l^{(\tau)} \right), \tag{3}$$

implying that subsequent updates do not significantly alter the representations learned on task $\tau$.

## 4  Methodology

While GPM has proven effective in centralized continual learning, extending it to federated settings introduces both optimization and privacy challenges. FOT (Bakman et al., 2024) offers an early adaptation by having clients share layer-wise intermediate activations, which the server uses to extract core bases. However, this approach raises significant privacy concerns, as such activations can be exploited in gradient inversion attacks (Geiping et al., 2020; Chen & Vikalo, 2024). It also introduces substantial communication overhead due to the high dimensionality of the transmitted activations.

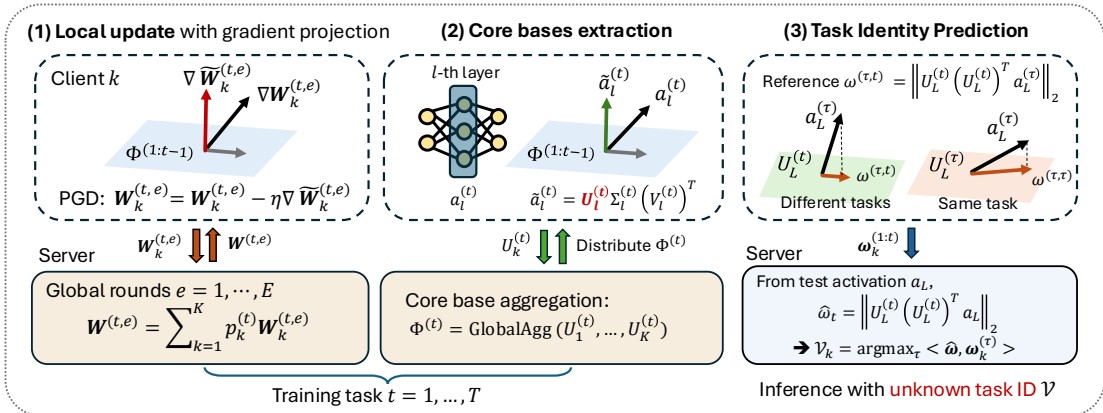

Figure 1: **Overview of FedProTIP.** (1) Clients apply projected gradient descent; the server aggregates updates. (2) Clients extract core bases via SVD; the server merges them into a global subspace. (3) At inference, task identity is predicted by comparing test relevance vectors to stored task references.

FOT performs standard local training on client devices and applies orthogonal projections to the global model update $\Delta \mathbf{W}^{(t)} = \sum_{k=1}^{K} p_k^{(t)} \Delta \mathbf{W}_k^{(t)}$ to mitigate feature interference across tasks. However, since local models are not trained with orthogonal constraints, this mismatch can lead to significant performance degradation under heterogeneous client data. Moreover, like most GPM-based methods, FOT assumes task identities are known during inference, which is unrealistic in many real-world deployments. In contrast, FedProTIP avoids both task ID reliance and the collection of intermediate activations, yet delivers strong performance under task-agnostic inference.

## 4.1 Local Training with Gradient Projection

As previously discussed, projecting only the aggregated global update may fail to constrain client-specific optimization trajectories under heterogeneous data. Instead, FedProTIP applies projected gradient descent (PGD) locally on each client and training batch. For task $t$ and global round $e \in \{1, \ldots, E\}$, client $k$ initializes $\mathbf{W}_k^{(t,e,0)} \leftarrow \mathbf{W}^{(t,e-1)}$ and performs $S$ local updates indexed by $s \in \{0, \ldots, S-1\}$. Let $\mathbf{\Phi}^{(1:t-1)}$ denote the matrix of core bases from earlier tasks and define the projection operator $\mathbf{P}^{(t)} \triangleq \mathbf{I} - \mathbf{\Phi}^{(1:t-1)} (\mathbf{\Phi}^{(1:t-1)})^{\top}$. FedProTIP then performs the projected update

$$\nabla \widetilde{\mathbf{W}}_k^{(t,e,s)} = \mathbf{P}^{(t)} \nabla \mathbf{W}_k^{(t,e,s)}, \tag{4}$$

$$\mathbf{W}_k^{(t,e,s+1)} = \mathbf{W}_k^{(t,e,s)} - \eta \nabla \widetilde{\mathbf{W}}_k^{(t,e,s)}, \tag{5}$$

where $\nabla \mathbf{W}_k^{(t,e,s)}$ denotes the stochastic gradient computed from client $k$'s local mini-batch at step $(t, e, s)$. (Projection is applied layer-wise; the layer index $l$ is omitted from subscripts for the sake of simplicity.) The operation in Eq. 4 removes gradient components aligned with past task subspaces, thereby reducing interference with prior knowledge and mitigating catastrophic forgetting. We empirically verify the effectiveness of this local projection mechanism in Section 6.3.

This design enforces the constraint $(\mathbf{\Phi}^{(1:t-1)})^{\top} (\mathbf{W}_k^{(t,e,s+1)} - \mathbf{W}_k^{(t,e,s)}) = \mathbf{0}$ at each local step, since $\mathbf{W}_k^{(t,e,s+1)} - \mathbf{W}_k^{(t,e,s)} = -\eta \mathbf{P}^{(t)} \nabla \mathbf{W}_k^{(t,e,s)}$ and $(\mathbf{\Phi}^{(1:t-1)})^{\top} \mathbf{P}^{(t)} = \mathbf{0}$. Summing over $s = 0, \ldots, S-1$ yields $(\mathbf{\Phi}^{(1:t-1)})^{\top} (\mathbf{W}_k^{(t,e,S)} - \mathbf{W}^{(t,e-1)}) = \mathbf{0}$, so each client update remains confined to the orthogonal complement of the past-task subspace.

## 4.2 Extracting Local Core Bases

After $S$ projected local updates within round $e$, client $k$ obtains the local iterate $\mathbf{W}_k^{(t,e,S)}$ and sends it to the server for aggregation. The server forms the updated global model

$$\mathbf{W}^{(t,e)} \triangleq \sum_{k=1}^{K} p_k^{(t)} \mathbf{W}_k^{(t,e,S)} \tag{6}$$

---

**Algorithm 1:** FedProTIP Training Procedure

---

**Input:** $K$ clients, $T$ tasks, the number of global rounds $E$, local datasets $\cup_{k \in [K]} \mathcal{D}_k^{(t)}$.

**Output:** The global model $\mathbf{W}^{(T)}$, stored bases $\mathbf{\Phi}^{(1:T)}$, references $(\boldsymbol{\omega}_k^{(t)})_{\forall k \in [K], t \in [T]}$.

1   **Initialization**: Broadcast $\mathbf{W}^{(0)}$ to all clients, $\mathbf{\Phi}^{(0)} \leftarrow \emptyset$;

2   **for** $t = 1, \dots, T$ **do**

3     $\mathbf{W}^{(t,0)} \leftarrow \mathbf{W}^{(t-1)}$;

4     **for** $e = 1, \dots, E$ **do**

5       **for** $k \in [K]$ **do**

6         $\mathbf{W}_k^{(t,e)} \leftarrow \mathbf{PGD}(\mathbf{W}^{(t,e-1)}, \mathcal{D}_k^{(t)}, \mathbf{\Phi}^{(0:t-1)})$; /* Following Eqs. (4)-(5)     */

7         Send $\mathbf{W}_k^{(t,e)}$ to the server;

8       **end**

9       $\mathbf{W}^{(t,e)} \leftarrow \sum_{k=1}^{K} p_k^{(t)} \mathbf{W}_k^{(t,e)}$;

10     **end**

11     **for** $k \in [K]$ **do**

12       $\mathbf{U}_k^{(t)}, \mathbf{a}_{L,k}^{(t)} \leftarrow \mathbf{ExtractBases}(\mathbf{W}^{(t,E)}, \mathcal{D}_k^{(t)}, \boldsymbol{\epsilon})$; /* Extract bases (Sec 4.2)     */

13       $\boldsymbol{\omega}_k^{(1:t)} \leftarrow \mathbf{UpdateReference}(\mathbf{U}_k^{(t)}, \mathbf{a}_{L,k}^{(t)}, \boldsymbol{\omega}_k^{(1:t-1)})$; /* Following Eq. (11)     */

14       Send $\mathbf{U}_k^{(t)}, \boldsymbol{\omega}_k^{(1:t)}$ to the server

15     **end**

16     $\mathbf{\Phi}^{(t)} \leftarrow \mathbf{GlobalAggregate}(\mathbf{U}_1^{(t)}, \dots, \mathbf{U}_K^{(t)})$; $\mathbf{W}^{(t)} \leftarrow \mathbf{W}^{(t,E)}$ /* Following Eq. (10)     */

17   **end**

---

and broadcasts it to all clients. After completing $E$ global rounds for task $t$, the resulting task-specific global model is denoted $\mathbf{W}^{(t)} \triangleq \mathbf{W}^{(t,E)}$, which is then used for local core basis extraction. Following the GPM strategy (Saha et al., 2021), each client $k$ samples $m$ examples from its local dataset $\mathcal{D}_k^{(t)}$, feeds them through the model $\mathbf{W}^{(t)}$, and collects layer-wise intermediate activations. Let $\mathbf{A}_l^{(t)} \in \mathbb{R}^{d_l \times m}$ denote the resulting activation matrix at layer $l$. Directly storing and decomposing $\mathbf{A}_l^{(t)}$ can be expensive when $m$ is large, while its effective rank is typically much smaller due to strong correlations among activation columns. Since core-basis extraction requires only the dominant singular subspace, we form a compact sketch by uniformly sampling a subset of $m_s \ll m$ columns, yielding $\mathbf{a}_l^{(t)} \in \mathbb{R}^{d_l \times m_s}$. This random activation sampling substantially reduces storage and communication overhead while retaining the dominant directions needed to estimate the task subspace. We evaluate the effect of sampling dimension in Appendix D.2. The sampled activations are then projected onto the orthogonal complement of the previously learned feature subspace by subtracting their component along the existing bases:

$$\tilde{\mathbf{a}}_l^{(t)} = \mathbf{a}_l^{(t)} - \mathbf{\Phi}^{(1:t-1)} \left( \mathbf{\Phi}^{(1:t-1)} \right)^\top \mathbf{a}_l^{(t)}. \tag{7}$$

The projected activations $\tilde{\mathbf{a}}_l^{(t)}$ are then decomposed using singular value decomposition (SVD),

$$\tilde{\mathbf{a}}_l^{(t)} = \mathbf{U}_l^{(t)} \mathbf{\Sigma}_l^{(t)} \left( \mathbf{V}_l^{(t)} \right)^\top, \tag{8}$$

where $\mathbf{U}_l^{(t)} \in \mathbb{R}^{d_l \times d_l}$ is a unitary matrix and $\mathbf{\Sigma}_l^{(t)} \in \mathbb{R}^{d_l \times m_s}$ is a diagonal matrix of singular values.

To extract the task-relevant bases, we select the smallest rank $r_l$ such that the retained singular values capture at least fraction $\epsilon_l$ of the total singular-value mass,

$$r_l = \min\left\{ r \in \mathbb{N} : \frac{\sum_{i=1}^{r} \sigma_{l,i}}{\sum_{i=1}^{\min(d_l, m_s)} \sigma_{l,i}} \geq \epsilon_l \right\}, \tag{9}$$

where $\sigma_{l,i}$ are the singular values of $\tilde{a}_l^{(t)}$ in descending order and $\epsilon_l \in (0, 1]$ is a layer-specific relative threshold. The resulting layer-wise core bases are

$$\mathbf{U}_l^{(t)} \leftarrow \mathbf{U}_l^{(t)}[\cdot, 1:r_l], \qquad l = 1, \dots, L.$$

Finally, each client $k$ sends its extracted local core bases $\{U_{l,k}^{(t)}\}_{l=1}^{L}$ to the server for aggregation. When stronger privacy protection is desired, these bases can be replaced by randomized sketches compatible with secure aggregation; see Section F.2.

### 4.3 Updating the Global Feature Subspace

The server collects core bases $\mathbf{U}_k^{(t)}$ from participating clients and integrates them into the global feature subspace by removing redundant components. Aggregation is initialized by setting $\mathbf{\Phi}_l^{(t)} \leftarrow \mathbf{U}_{l,1}^{(t)}$ for each layer $l$, using the bases received from the first client. The server then iteratively incorporates bases from the remaining clients by forming the residual component of each local basis with respect to the current aggregated subspace and appending it:

$$\mathbf{\Phi}_l^{(t)} \leftarrow \left[ \mathbf{\Phi}_l^{(t)}, \mathbf{U}_{l,k}^{(t)} - \mathbf{\Phi}_l^{(t)} \left( \mathbf{\Phi}_l^{(t)} \right)^\top \mathbf{U}_{l,k}^{(t)} \right], \qquad k = 2, \dots, K. \tag{10}$$

This orthogonal appending step removes redundancy without discarding any new client-specific directions: the aggregated global basis spans the union of the client-transmitted local bases. A formal proof is given in Appendix C. The appended vectors are subsequently orthonormalized to maintain an orthonormal basis for the aggregated subspace. Following aggregation, the updated global bases $\mathbf{\Phi}^{(t)}$ are broadcast to clients and used in the next task's training phase, as described in Section 4.1. Because each $\mathbf{U}_{l,k}^{(t)}$ is extracted from activations already projected against $\mathbf{\Phi}_l^{(1:t-1)}$, the new aggregated basis $\mathbf{\Phi}_l^{(t)}$ contributes only previously unseen directions. Hence, the full updated subspace is obtained by augmenting $\mathbf{\Phi}_l^{(1:t-1)}$ with $\mathbf{\Phi}_l^{(t)}$.

### 4.4 Task Identification via Subspace Relevance

In continual learning, the feature extractor is fine-tuned across sequential tasks, while the decision head expands as new tasks are introduced. For example, in class-incremental settings, the dimensionality of the softmax output layer grows with the number of classes. Prior works (Saha et al., 2021; Bakman et al., 2024) assume that the task identity $\tau$ is known at test time so that predictions can be routed through the corresponding decision head $f_\tau(\cdot)$. In practice, however, this assumption is often unrealistic because task labels are typically unavailable during deployment (Kim et al., 2022b).

To address this challenge, FedProTIP introduces a task identification mechanism based on two key concepts: subspace relevance and reference vectors. Subspace relevance quantifies how strongly a representation aligns with the feature subspace associated with each learned task. Reference vectors capture the characteristic relevance patterns observed for previously learned tasks. As illustrated in Fig. 1, each client constructs these reference vectors from training data by measuring how its final-layer activations project onto the task subspaces. At test time, the model computes a relevance vector for a new input and compares it with the stored reference vectors to determine the most likely task identity.

**Client-side reference vector computation.** During local training on task $\tau$, each client records layer-wise intermediate activation vectors, denoted by $\mathbf{a}_l^{(\tau)}$. For task identification we use the input to the final layer, $\mathbf{a}_L^{(\tau)}$. Let $\mathbf{U}_L^{(t)}$ denote the global task-specific core bases at the final layer for each task $t \in \{1, \dots, T\}$. After completing $T$ tasks, each client forms, for every $\tau \leq T$, a reference vector $\boldsymbol{\omega}^{(\tau)} = [\omega^{(\tau,1)}, \dots, \omega^{(\tau,T)}] \in \mathbb{R}^T$, where the $t$-th entry is the projection magnitude of $\mathbf{a}_L^{(\tau)}$ onto the task-$t$ subspace:

$$\omega^{(\tau,t)} \triangleq \left\| \mathbf{U}_L^{(t)} \left( \mathbf{U}_L^{(t)} \right)^\top \mathbf{a}_L^{(\tau)} \right\|_2, \qquad \forall \tau, t \in \{1, \dots, T\}. \tag{11}$$

Thus, $\omega^{(\tau,t)}$ quantifies how strongly task $\tau$'s representation aligns with the subspace learned for task $t$. As noted in Section 4.2, this value is typically small for $\tau < t$ in practice because later task subspaces are constructed from activations orthogonalized with respect to previously learned representations. Each client $k$ stores the set of reference vectors $\{\boldsymbol{\omega}_k^{(\tau)}\}_{\tau=1}^T$ and transmits them to the server for task identification during deployment.

**Test-time task identification.** Given a test sample, the model computes the final-layer activation $\mathbf{a}_L^{\text{te}}$ and forms a subspace relevance vector $\hat{\boldsymbol{\omega}} = [\hat{\omega}^{(1)}, \dots, \hat{\omega}^{(T)}] \in \mathbb{R}^T$ with $\hat{\omega}^{(t)} \triangleq \|\mathbf{U}_L^{(t)}(\mathbf{U}_L^{(t)})^\top \mathbf{a}_L^{\text{te}}\|_2, \forall t \in \{1, \dots, T\}$. The server compares this subspace relevance vector with the stored reference vectors using cosine similarity,

$$\mathcal{S}_k^{(\tau)} = \frac{\hat{\boldsymbol{\omega}} \cdot \boldsymbol{\omega}_k^{(\tau)}}{\|\hat{\boldsymbol{\omega}}\| \, \|\boldsymbol{\omega}_k^{(\tau)}\|}, \quad \forall k \in [K], \ \tau \in \{1, \dots, T\}. \tag{12}$$

For each client index $k$, the task with the highest similarity is selected, $\mathcal{V}_k = \arg\max_\tau \mathcal{S}_k^{(\tau)}$, and the server determines the final task identity by majority vote across clients. Because the subspace relevance vectors $\hat{\boldsymbol{\omega}} \in \mathbb{R}^T$ are low-dimensional, this procedure incurs negligible computational overhead. In large-scale FL systems, task identification can be efficiently approximated using only a representative subset of clients.

TIP relies on the standard continual learning assumption that each task draws from a distinct distribution $\mathcal{D}^{(t)}$. Under this assumption, task-specific training can induce distinguishable activation patterns, and the projected task subspaces can capture directions useful for task identification. When this assumption is violated, e.g., when identical inputs are reused with different labeling semantics, TIP may fail to distinguish tasks. This limitation is outside the scope of the current framework.

## 5 Theoretical Analysis

In this section, we analyze the convergence of FedProTIP. Since TIP is an inference-time mechanism and does not alter the projected local training recursion, the analysis focuses on the replay-free training component. We study two quantities: (i) convergence on the current task during training of that task, and (ii) cumulative loss increase on previously learned tasks due to subsequent training. Full assumptions and proofs are provided in Appendix A.

Our bounds depend on two geometric quantities induced by the learned subspaces. The first is a projected-gradient adequacy coefficient $\rho_t \in (0, 1]$, which measures how much current-task descent remains available after projection. The second is an interference coefficient $\beta_\tau^{(t-1)} \in [0, 1]$, which measures how much old-task gradient energy remains inside the admissible update space while learning task $t$.

**Theorem 1** (Task-wise convergence). *Let $\Delta_t := L^{(t)}(W^{(t,0)}) - L^{(t)\star}$, where $L^{(t)\star} := \inf_W L^{(t)}(W)$. Under the assumptions stated in Appendix A, the iterates generated while learning task $t$ satisfy*

$$\frac{1}{E_t} \sum_{e=1}^{E_t} \mathbb{E}\big\|\nabla L^{(t)}(W^{(t,e-1)})\big\|^2 \leq \frac{2\Delta_t}{\rho_t E_t S_t \eta_t} + \frac{L\eta_t S_t G^2}{\rho_t}\left(1 + \frac{L\eta_t S_t}{3\rho_t}\right). \tag{13}$$

**Theorem 2** (Cumulative forgetting). *For every pair of tasks $\tau < t$, the cumulative loss increase of task $\tau$ after all later tasks have been learned satisfies*

$$\mathbb{E}\big[L^{(\tau)}(W^{(T)})\big] - \mathbb{E}\big[L^{(\tau)}(W^{(\tau)})\big] \leq G^2 \sum_{t=\tau+1}^{T} E_t \left(\beta_\tau^{(t-1)} S_t \eta_t + \frac{L}{2} S_t^2 \eta_t^2\right). \tag{14}$$

**Corollary 1** (Stability–plasticity trade-off). *If $\eta_t = \frac{1}{LS_t\sqrt{E_t}}$, then*

$$\frac{1}{E_t} \sum_{e=1}^{E_t} \mathbb{E}\big\|\nabla L^{(t)}(W^{(t,e-1)})\big\|^2 \leq \frac{2L\Delta_t}{\rho_t\sqrt{E_t}} + \frac{G^2}{\rho_t\sqrt{E_t}}\left(1 + \frac{1}{3\rho_t\sqrt{E_t}}\right), \tag{15}$$

*so the current-task stationarity measure decays at rate $\mathcal{O}(1/\sqrt{E_t})$. Moreover, if $E_t \equiv E$, $S_t \equiv S$, and $\eta_t \equiv \eta$ across tasks, then the average loss-based forgetting satisfies*

$$FT_{\text{loss}}(T) \leq \frac{TG^2\bar{\beta}_T}{2L}\sqrt{E} + \frac{TG^2}{4L}, \tag{16}$$

*where $\bar{\beta}_T := \frac{2}{T(T-1)} \sum_{t=2}^{T} \sum_{\tau=1}^{t-1} \beta_\tau^{(t-1)}$.*

**Remarks.** Theorems 1 and 2 characterize the stability–plasticity trade-off through $\rho_t$ and $\beta_\tau^{(t-1)}$. The quantity $\rho_t$ captures current-task plasticity, since a larger $\rho_t$ yields a tighter task-wise convergence bound by preserving more descent directions after projection. The quantities $\beta_\tau^{(t-1)}$ and $\bar{\beta}_T$ capture cross-task interference, so smaller values imply better stability and smaller cumulative loss increase. Under the canonical step-size schedule, the task-wise convergence bound scales as $\mathcal{O}(1/(\rho_t\sqrt{E_t}))$ up to lower-order terms, whereas the forgetting bound scales as $\mathcal{O}(T\bar{\beta}_T\sqrt{E})$. Notably, the bounds are expressed in terms of $\rho_t$ and $\beta_\tau^{(t-1)}$ rather than the practitioner-facing threshold $\epsilon_l$. Fig. 2 bridges this gap empirically on 10-split CIFAR100: both $\hat{\rho}_t$ and $\hat{\bar{\beta}}$ decrease monotonically with $\epsilon_l$, indicating that the threshold provides an empirical handle on

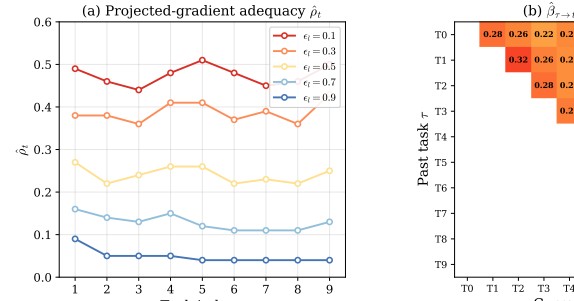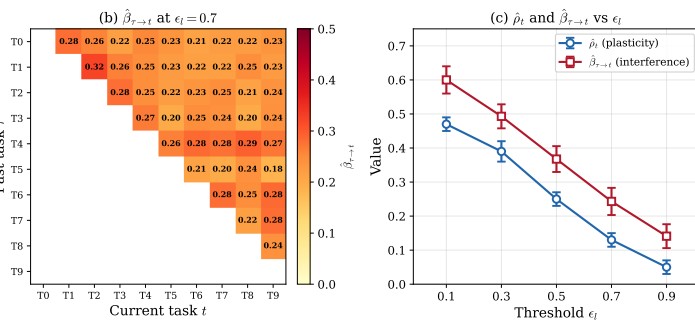

Figure 2: Empirical estimation of theoretical quantities on 10-split CIFAR100 ($\alpha = 0.5$). (a) Projected-gradient adequacy $\hat{\rho}_t$ across tasks for five thresholds. (b) Interference coefficient $\hat{\beta}_{\tau \to t}$ heatmap at threshold $\epsilon_l = 0.7$. (c) Both quantities as a function of $\epsilon_l$, averaged across tasks.

the stability–plasticity trade-off identified by the bounds. Concretely, increasing $\epsilon_l$ enlarges the protected subspace, which reduces $\hat{\beta}$ (less interference with past tasks) at the cost of lower $\hat{\rho}_t$ (less gradient energy available for the current task). Because these opposing effects partially offset one another, end-task accuracy remains largely insensitive to $\epsilon_l$, as observed in Fig. 4b.

# 6 Experiments

We evaluate FedProTIP on three standard continual learning benchmarks: CIFAR100 and ImageNet-R (Hendrycks et al., 2021) for class-incremental learning, and DomainNet (Peng et al., 2019) for domain-incremental learning. CIFAR100 is divided into 10 tasks with 10 classes each, while ImageNet-R is evaluated under both 10- and 20-task splits. For DomainNet, we follow the domain-incremental protocol in which tasks correspond to different visual domains while sharing the same label space. We compare FedProTIP against six representative baselines: FedAvg (McMahan et al., 2017), GLFC (Dong et al., 2022), LGA (Dong et al., 2023), TARGET (Zhang et al., 2023), FOT (Bakman et al., 2024), and LANDER (Tran et al., 2024). These include replay-based, generative, and projection-based approaches for federated continual learning. For FedProTIP, we use a common initial threshold $\epsilon_l = \epsilon$ for all layers and increase it by 0.001 at each task boundary as the default schedule. The effect of different threshold choices is examined separately in the ablation study. Following (Yurochkin et al., 2019), we simulate non-IID client distributions by sampling client data partitions from a Dirichlet distribution with concentration parameter $\alpha$, where smaller $\alpha$ corresponds to greater data heterogeneity. All methods use a ResNet-18 backbone pretrained on ImageNet-1K (He et al., 2016), following common practice in CL benchmarks, and are fine-tuned on each dataset. Additional backbone studies, including ResNets trained from scratch and Vision Transformers, are reported in Appendix D.1.

Following prior work (Chaudhry et al., 2018), we evaluate performance using two standard metrics: average accuracy (ACC) and forgetting (FT), defined as

$$\text{ACC} = \frac{1}{T} \sum_{t=1}^{T} \text{acc}_t^{(T)}, \quad \text{FT} = \frac{1}{T} \sum_{t=1}^{T-1} \left( \max_{i \in \{t, \dots, T-1\}} \text{acc}_t^{(i)} - \text{acc}_t^{(T)} \right), \tag{17}$$

where $\text{acc}_t^{(i)}$ denotes the accuracy on task $t$ after learning $i$ tasks, and $\text{acc}_t^{(T)}$ is the final accuracy on task $t$ after all $T$ tasks have been learned. Task identities are not provided during inference, consistent with the task-agnostic evaluation setting. Additional experimental details are provided in Appendix E.

## 6.1 Performance in Task-Agnostic and Task-Aware Settings

Fig. 3 reports average task accuracy ($y$-axis) as a function of the number of learned tasks ($x$-axis). In the task-agnostic setting, where the task identity of test samples is unknown (Fig. 3a), FedProTIP consistently outperforms all baselines across the entire task sequence. While several baselines achieve competitive accuracy when the task identity is provided at test time (Fig. 3b), their performance degrades substantially in the task-agnostic setting. In contrast, FedProTIP maintains strong accuracy across the task sequence by combining orthogonal gradient projection, which reduces cross-task interference during training, with task

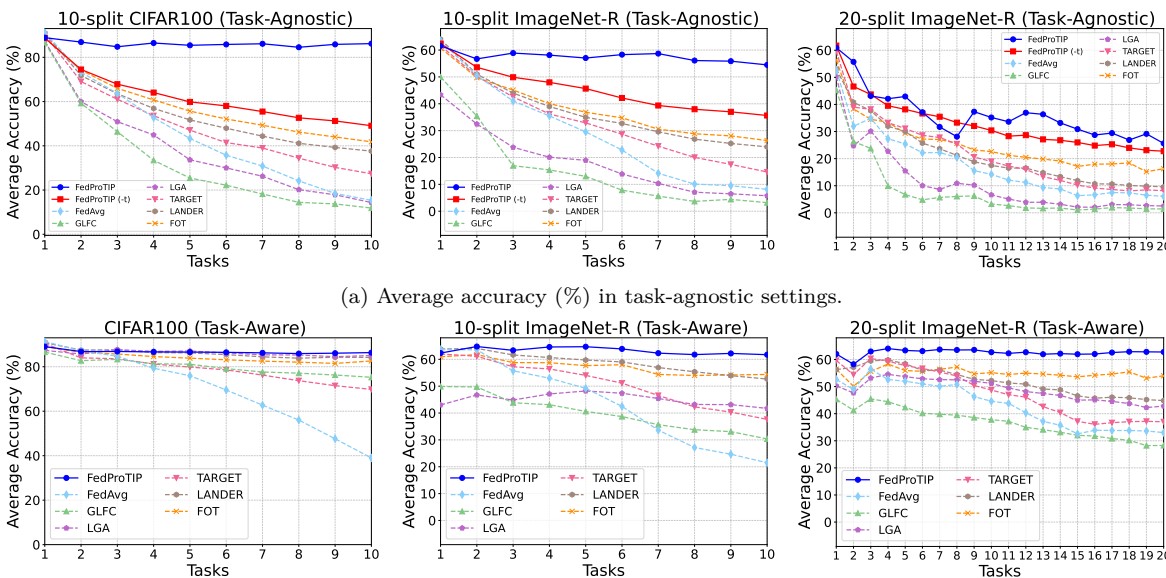

(a) Average accuracy (%) in task-agnostic settings.

(b) Average accuracy (%) in task-aware settings where true task-ID is provided during inference.

Figure 3: Average accuracy of class-incremental learning on three benchmarks. (a) Task-agnostic inference, where task identity is unknown. (b) Task-aware inference, where the true task ID is provided at test time.

Table 1: Accuracy (↑) and forgetting (↓) metrics (%) on 10-split CIFAR-100 and 6-split DomainNet across different heterogeneity levels (Dirichlet $\alpha$). **Bold** and underline indicate the best and second-best results, respectively. GLFC and LGA are incompatible with domain-incremental learning and are marked with $\star$. All results are reported under task-agnostic inference.

| Method | 10-Split CIFAR100 (Class-IL) | | | | | | 6-Split DomainNet (Domain-IL) | | | | | |
| | IID | | $\alpha = 0.5$ | | $\alpha = 0.2$ | | IID | | $\alpha = 0.5$ | | $\alpha = 0.2$ | |
| | ACC | FT | ACC | FT | ACC | FT | ACC | FT | ACC | FT | ACC | FT |
|---|---|---|---|---|---|---|---|---|---|---|---|---|
| FedAvg | 18.92 | 63.20 | 15.35 | 62.90 | 15.76 | 52.80 | 10.79 | 27.74 | 10.72 | 25.66 | 10.53 | 25.57 |
| GLFC | 14.07 | 69.17 | 11.86 | 68.20 | 10.33 | 63.98 | $\star$ | $\star$ | $\star$ | $\star$ | $\star$ | $\star$ |
| LGA | 14.93 | 72.06 | 14.35 | 71.09 | 11.67 | 65.82 | $\star$ | $\star$ | $\star$ | $\star$ | $\star$ | $\star$ |
| TARGET | 29.56 | 42.73 | 27.37 | 37.60 | 23.05 | 34.63 | 21.53 | 9.73 | 20.61 | 7.89 | 20.64 | 8.31 |
| LANDER | 39.09 | 9.27 | 37.59 | 10.21 | 23.56 | 13.28 | 21.88 | 8.90 | 21.59 | 10.27 | 22.11 | 8.59 |
| FOT | 46.86 | 21.11 | 41.80 | 20.86 | 34.65 | 18.09 | 24.59 | 8.85 | 24.13 | 8.44 | 23.84 | 8.33 |
| FedProTIP (-t) | 52.30 | 15.66 | 48.41 | 15.59 | 42.19 | 14.91 | **29.64** | 6.38 | **28.85** | 6.43 | **28.74** | 6.14 |
| **FedProTIP** | **87.94** | **1.30** | **86.00** | **0.83** | **81.94** | **1.35** | 27.60 | **2.89** | 25.30 | **3.76** | 25.98 | **2.88** |

identification that routes test samples to the appropriate output head. Even without task identification, the projection-only variant (FedProTIP (-t)) still outperforms FOT, highlighting the effectiveness of the proposed subspace-based projection mechanism and the globally aggregated core bases. The gap between FedProTIP and FedProTIP(-t) is closely tied to task routing quality. As shown in Fig. 4a, TIP achieves $\geq 0.978$ across all tasks on 10-split CIFAR100, remains above 0.87 on 10-split ImageNet-R, and stays above 0.67 on 6-split DomainNet despite the shared label space. Although task identification becomes less reliable in the more challenging 20-split ImageNet-R setting, where each task contains fewer classes and examples to define distinctive subspaces, both variants of FedProTIP remain superior to all baselines, demonstrating robust scalability as the number of tasks increases. A full per-task breakdown of TIP routing accuracy is provided in Table 14 (Appendix D).

## 6.2 Robustness under Data Heterogeneity and Forgetting

**Data heterogeneity.** As shown in Table 1, FedProTIP consistently outperforms all baselines across various values of the Dirichlet concentration parameter $\alpha$, which controls the degree of data heterogeneity. As $\alpha$ decreases and client data distributions become increasingly non-IID, client drift (Karimireddy et al., 2020) typically becomes more pronounced and exacerbates catastrophic forgetting. Despite this challenge,

Table 2: Accuracy (↑) and forgetting (↓) metrics (%) computed in the experiments on 5-split, 10-split, and 20-split ImageNet-R. **Bold** and underline indicate the best and the second-best methods, respectively.

| Method | 5-Split ImageNet-R | | | | 10-Split ImageNet-R | | | | 20-Split ImageNet-R | | | |
|---|---|---|---|---|---|---|---|---|---|---|---|---|
| | IID | | $\alpha = 0.5$ | | IID | | $\alpha = 0.5$ | | IID | | $\alpha = 0.5$ | |
| | ACC | FT | ACC | FT | ACC | FT | ACC | FT | ACC | FT | ACC | FT |
| FedAvg | 22.70 | 37.11 | 22.22 | 36.26 | 8.74 | 43.84 | 8.15 | 41.14 | 9.77 | 43.18 | 6.08 | 31.75 |
| GLFC | 7.26 | 16.99 | 7.47 | 17.12 | 3.34 | 29.88 | 3.18 | 29.80 | 2.12 | 36.12 | 1.43 | 30.40 |
| LGA | 8.33 | 21.13 | 7.38 | 19.91 | 5.84 | 36.41 | 5.76 | 35.05 | 3.32 | 43.29 | 2.52 | 40.76 |
| TARGET | 40.95 | 14.43 | 37.71 | 14.89 | 17.64 | 25.83 | 14.60 | 23.52 | 9.77 | 29.87 | 8.18 | 24.63 |
| LANDER | 35.50 | **1.45** | 36.83 | **1.46** | 24.53 | 5.39 | 23.96 | 3.10 | 12.23 | **10.33** | 8.73 | **8.00** |
| FOT | 39.77 | 13.43 | 38.58 | 13.24 | 23.68 | 14.61 | 26.31 | 15.52 | 22.50 | 16.08 | 16.27 | 13.26 |
| FedProTIP (-t) | 50.00 | 6.26 | 46.99 | 8.03 | 41.35 | 8.80 | 35.64 | 8.65 | 31.43 | 10.37 | 22.75 | 10.97 |
| **FedProTIP** | **55.65** | 3.36 | **54.49** | 6.03 | **52.68** | 10.34 | **54.48** | 7.48 | **34.80** | 12.03 | **25.62** | 12.21 |

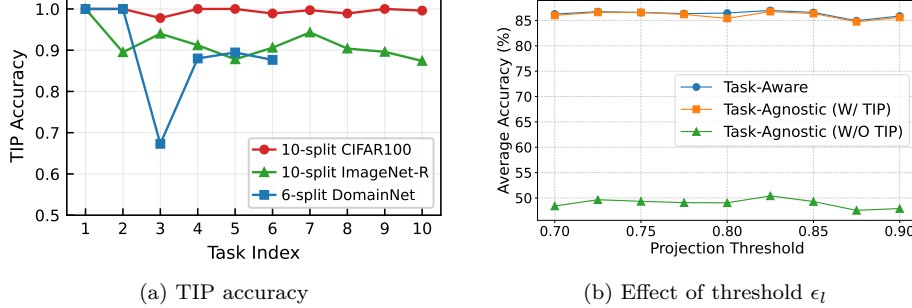

(a) TIP accuracy                 (b) Effect of threshold $\epsilon_l$

Figure 4: (a) Task identity prediction accuracy across tasks ($\alpha = 0.5$). (b) Effect of projection threshold $\epsilon_l$ on FedProTIP accuracy on 10-split CIFAR100 ($\alpha = 0.5$).

FedProTIP remains robust. For example, on CIFAR100, the accuracy of competing methods such as FOT and LANDER drops by 12% and 15%, respectively, when moving from IID partitions to $\alpha = 0.2$, whereas FedProTIP exhibits only a 6% decline. At the same time, it maintains near-zero forgetting, indicating strong resilience to heterogeneous client updates. We further evaluate robustness on DomainNet in the domain-incremental setting, where all tasks share the same 345-class label space. Under task-agnostic inference, FedProTIP(-t) uses a single shared classifier that benefits from cross-domain knowledge transfer, and it consistently achieves the highest accuracy across heterogeneity levels. The full FedProTIP model instead maintains separate task-specific classifiers routed by TIP, which isolates per-domain representations and achieves the lowest forgetting, but at the cost of occasional routing errors that slightly reduce accuracy. This reflects a trade-off specific to the domain-incremental regime: when tasks share the same label space, a shared head can be preferable for accuracy, while task-specific heads can better preserve domain-specific representations (Fig. 4a).

**Catastrophic forgetting.** Table 2 reports results on ImageNet-R under 5-, 10-, and 20-task splits. As the number of tasks increases, forgetting accumulates and overall accuracy decreases for all methods, as reflected in the higher forgetting values observed in larger splits. While LANDER often achieves the lowest forgetting, it does so at the expense of substantially lower accuracy. In contrast, FedProTIP achieves a stronger balance between accuracy and forgetting, outperforming the second-best method (FOT) by 8%–28% in accuracy while maintaining competitive forgetting across all splits. Even in the 20-task setting, FedProTIP sustains 25%–35% accuracy, substantially higher than competing methods. These results indicate that FedProTIP degrades more gracefully than prior methods as the number of tasks increases, an important property for practical continual learning systems.

### 6.3 Local vs. Global Gradient Projection

To evaluate the impact of using local versus global projection in challenging federated settings, we measure test accuracy under progressively increasing client heterogeneity, controlled by a Dirichlet parameter $\alpha \in \{0.05, 0.1, 0.2\}$. As shown in Figs. 5a and 5b, local projection consistently outperforms global projection in

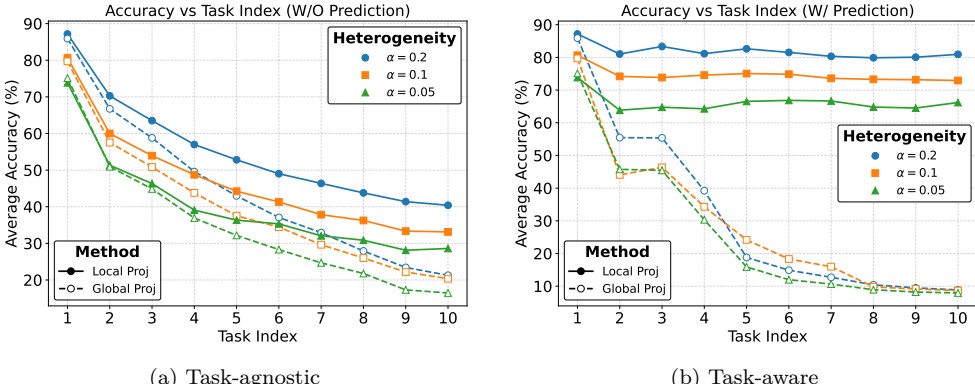

(a) Task-agnostic

(b) Task-aware

Figure 5: Impact of gradient projection strategies. Local vs. global projection on 10-split CIFAR100 under $\alpha \in \{0.2, 0.1, 0.05\}$.

Table 3: Accuracy ($\uparrow$) and forgetting ($\downarrow$) on 10-split CIFAR-100 under different numbers of clients (5, 10, 20).

| Method | 10-Split CIFAR100 ($\alpha = 0.5$) | | | | | |
| --- | --- | --- | --- | --- | --- | --- |
| | 5 | | 10 | | 20 | |
| | ACC | FT | ACC | FT | ACC | FT |
| GLFC | 11.86 | 68.20 | 9.55 | 61.04 | 7.45 | 51.42 |
| LGA | 14.35 | 71.09 | 12.24 | 69.06 | 13.36 | 63.58 |
| TARGET | 27.37 | 37.60 | 23.28 | 40.65 | 22.41 | 43.27 |
| LANDER | 37.59 | 10.21 | 26.60 | 2.16 | 23.42 | 6.13 |
| FOT | 41.80 | 20.86 | 39.73 | 13.08 | 37.35 | 13.66 |
| FedProTIP (-t) | 48.41 | 15.59 | 41.33 | 10.70 | 40.06 | 9.55 |
| **FedProTIP** | **86.00** | **0.83** | **81.34** | **0.59** | **81.10** | **0.28** |

Table 4: Asymptotic per-task overhead beyond standard federated SGD. Here, $d_l$ denotes layer width, $r_l$ the retained rank, $s_l$ the FOT sketch dimension ($s_l = 5d_l$), $S$ the number of local steps, $|\theta_G|$ the number of generator parameters, $C$ the number of classes, and $d_e$ embedding dimension.

| Method | Local | Comm. | Memory |
| --- | --- | --- | --- |
| FedProTIP | $S\sum d_l r_l$ | $\sum d_l r_l$ | $\sum d_l r_l$ |
| FOT | — | $\sum d_l s_l$ | $\sum d_l r_l$ |
| TARGET | $|\theta_G|$ | $|\theta_G|$ | $|\theta_G|$ |
| LANDER | $|\theta_G|+Cd_e$ | $|\theta_G|$ | $|\theta_G|+Cd_e$ |

mitigating catastrophic forgetting. In the global baseline, local updates remain unconstrained during client training, allowing gradients to drift into previously learned subspaces before aggregation. This interference accumulates across local steps and becomes more severe as client distributions grow increasingly heterogeneous, making projection applied only after aggregation insufficient. By contrast, FedProTIP enforces projection throughout local optimization, suppressing cross-task interference early in training and leading to improved robustness even under extreme heterogeneity ($\alpha = 0.05$). Finally, although TIP further improves overall class-incremental performance, global projection remains inferior because it does not preserve task-specific orthogonal subspaces as effectively, which in turn degrades task identification.

### 6.4 Ablation Studies

**Varying number of clients.** To evaluate how FedProTIP scales with the number of clients in the federated system, we conduct experiments with 5, 10, and 20 clients. To keep the expected number of participating clients per communication round constant, we set the client sampling rates to 1, 0.5, and 0.25, respectively. As shown in Table 3, FedProTIP consistently outperforms competing methods across all configurations. Increasing the number of clients generally leads to performance degradation for all FCL methods, as it typically increases data heterogeneity and reduces local data diversity. Nevertheless, FedProTIP remains robust in these settings. The use of orthogonal gradient projection reduces cross-task interference during local updates, which helps mitigate forgetting and preserve accuracy.

**Effect of the projection threshold.** FedProTIP extracts core bases from layer activations using a layer-wise threshold $\epsilon_l$, which determines how many principal directions are retained at each layer. We assess sensitivity to this parameter by varying $\epsilon_l \in [0.7, 0.9]$ across all layers on CIFAR100 (Fig. 4b). Results show that FedProTIP is largely insensitive to the threshold choice, maintaining stable accuracy across this range. Very high thresholds preserve more directions from previous tasks, improving stability but reducing plasticity because fewer orthogonal directions remain available for new tasks. This reflects the standard stability–plasticity trade-off: moderate thresholds provide a good balance between preserving prior knowledge

Table 5: Comparison with federated variants of task-agnostic inference methods. $\Delta$ values denote performance gains when combined with FCL methods.

| Method | Task-Agnostic | | Task-Aware | |
|---|---|---|---|---|
| | $\text{ACC}_\Delta$ | $\text{FT}_\Delta$ | ACC | FT |
| Fed+PEC | 19.56 | 12.18 | 50.04 | 0.00 |
| Fed+SCR | 34.26 | 38.22 | – | – |
| Tar+LODE | $29.87_{\uparrow 2.50}$ | $44.09_{\uparrow 6.48}$ | 69.24 | 15.96 |
| FedProTIP | $86.00_{\uparrow 37.19}$ | $0.83_{\downarrow 14.75}$ | 86.26 | 0.96 |

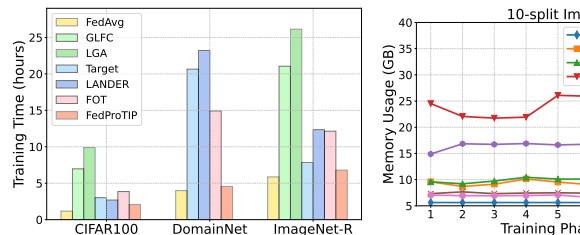

Figure 6: Training efficiency comparison: (left) training time (hours), (right) average GPU memory usage.

and adapting to new tasks. Additional experiments on DomainNet and ImageNet-R (Appendix D.4) confirm this behavior and show consistent robustness across datasets.

**Impact of task prediction strategies.** We investigate whether task-agnostic continual learning methods originally developed for centralized settings can be adapted to federated scenarios. Specifically, we consider the replay-free PEC (Zając et al., 2024), which assigns a separate classifier to each class, and the replay-based SCR (Mai et al., 2021), which uses a nearest-class-mean classifier. For SCR, we average class prototypes across clients at inference time while keeping replay data local, and perform model aggregation using FedAvg. As shown in Table 5, both methods yield significantly lower accuracy under task-agnostic inference. While SCR outperforms PEC, it relies on clients retaining local data, which may violate privacy constraints. We also evaluate LODE (Liang & Li, 2023b), which decouples intra- and inter-task losses to implicitly support task-agnostic inference. Applied to the generative replay method TARGET, LODE offers only marginal gains (+2.50%), suggesting that naive loss decoupling is insufficient for robust performance in federated settings.

### 6.5 Training Times, Memory Usage, and Communication Cost

**Asymptotic complexity.** FedProTIP's per-step overhead consists of gradient projection at $\mathcal{O}(d_l r_l)$ operations per layer. Its task-level overhead comes from basis extraction via randomized SVD on sampled activation matrices $\mathbf{a}_l^{(t)} \in \mathbb{R}^{d_l \times m_s}$ with $m_s \ll m$. Persistent per-task memory scales as $\mathcal{O}(\sum_l d_l r_l^{(t)})$, storing only low-rank core bases $U_l^{(t)} \in \mathbb{R}^{d_l \times r_l^{(t)}}$, which is substantially smaller than replay-based storage $\mathcal{O}(N_{\text{replay}} CHW)$ and typically decreases in later tasks as fewer novel directions are retained. Per-layer communication cost is $\mathcal{O}(d_l r_l)$ per client, compared to $\mathcal{O}(d_l s_l)$ for FOT's randomized activation sketches. A summary comparing asymptotic computation, communication, and memory across methods is provided in Table 4. We note that the projection cost scales with network depth and layer width.

**Empirical measurements.** Across all datasets, FedProTIP is the fastest method among the continual learning baselines, second only to FedAvg (which does not include a continual-learning mechanism). On high-resolution datasets such as DomainNet, FedProTIP trains up to $5\times$ faster than generative baselines such as TARGET and LANDER. FedProTIP also attains the lowest peak GPU memory among FCL methods (Figs. 6 and 7), since it does not maintain replay batches or generative models. As summarized in Table 9 in the Appendix, for 10-split CIFAR100 FOT incurs a fixed 48 MB per task, while FedProTIP typically communicates less than 10 MB in early tasks and even less in later ones, reducing communication by roughly an order of magnitude.

## 7 Conclusion

We proposed FedProTIP, a federated continual learning framework that leverages gradient projection to reduce feature interference and mitigate catastrophic forgetting. Unlike many prior FCL methods, FedProTIP requires neither storing past data nor training generative models for rehearsal. The method extracts core feature subspaces via memory-efficient randomized SVD and uses them for task identification, enabling improved alignment between test inputs and decision layers. Extensive experiments across three benchmark datasets show that FedProTIP consistently outperforms existing methods while maintaining lower computational and communication overhead.

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

# A    Theoretical Results

We analyze the training dynamics of FedProTIP *without* the task-identity prediction (TIP) module. This restriction is deliberate: TIP is an inference-time mechanism built on top of the learned subspaces and does not alter the training recursion in Eqs. (20) and (21). Accordingly, the present section focuses on the projected local optimization procedure that distinguishes FedProTIP from prior replay-free FCL methods.

Following the decomposition used in recent continual federated learning analyses, we study two quantities separately: (i) convergence on the *current* task while that task is being trained, and (ii) cumulative loss increase on *previously learned* tasks caused by later training. We intentionally do *not* state a final bound on the task-agnostic test accuracy or on the average multi-task objective after all tasks. Such a theorem would require additional assumptions linking task-wise losses, classifier routing, and task-identity prediction. In the absence of those assumptions, a final end-to-end bound is too loose to be informative.

## A.1    Setup and notation

Consider $K$ clients and a sequence of $T$ tasks. Client $k$ receives local data $\mathcal{D}_k^{(t)}$ for task $t$, with aggregation weight $p_k^{(t)} \geq 0$ and $\sum_{k=1}^{K} p_k^{(t)} = 1$. The global loss of task $t$ is

$$L^{(t)}(\mathbf{W}) \triangleq \sum_{k=1}^{K} p_k^{(t)} L_k^{(t)}(\mathbf{W}), \qquad L_k^{(t)}(\mathbf{W}) \triangleq \mathbb{E}_{\xi \sim \mathcal{D}_k^{(t)}}\big[\ell(\mathbf{W}; \xi)\big]. \tag{18}$$

When task $t$ is trained, data from tasks $1, \ldots, t-1$ are unavailable.

After finishing task $t-1$, the server holds a global orthonormal basis matrix $\boldsymbol{\Phi}^{(1:t-1)} \in \mathbb{R}^{d \times r_{t-1}^{\mathrm{agg}}}$ obtained by aggregating the low-rank bases extracted from all clients using the orthogonal appending rule of Section 4.3. The corresponding orthogonal projector onto the admissible update space is

$$P^{(t-1)} \triangleq I - \boldsymbol{\Phi}^{(1:t-1)}\big(\boldsymbol{\Phi}^{(1:t-1)}\big)^{\top}. \tag{19}$$

FedProTIP then trains task $t$ by local projected gradient descent. At global round $e \in \{1, \ldots, E_t\}$, each client initializes $\mathbf{W}_k^{(t,e,0)} = \mathbf{W}^{(t,e-1)}$ with $\mathbf{W}^{(t,0)} = \mathbf{W}^{(t-1)}$, performs $S_t$ local projected steps,

$$\mathbf{g}_k^{(t,e,s)} \triangleq \nabla_{\mathbf{W}}\ell\big(\mathbf{W}_k^{(t,e,s)}; \xi_k^{(t,e,s)}\big), \qquad \widetilde{\mathbf{g}}_k^{(t,e,s)} \triangleq P^{(t-1)}\mathbf{g}_k^{(t,e,s)}, \tag{20}$$

$$\mathbf{W}_k^{(t,e,s+1)} = \mathbf{W}_k^{(t,e,s)} - \eta_t \widetilde{\mathbf{g}}_k^{(t,e,s)}, \qquad s = 0, \ldots, S_t - 1, \tag{21}$$

and the server averages the resulting local models,

$$\mathbf{W}^{(t,e)} = \sum_{k=1}^{K} p_k^{(t)} \mathbf{W}_k^{(t,e,S_t)}. \tag{22}$$

We write $\mathbf{W}^{(t)} \triangleq \mathbf{W}^{(t,E_t)}$ for the model after task $t$.

The key point is that projection is applied *before* local optimization is completed, not only after server aggregation. This is exactly the algorithmic feature that differentiates FedProTIP from FOT under heterogeneous client data.

## A.2    Assumptions

**Assumption 1** (*L*-smooth task losses)**.** *For every task $t$ and client $k$, the local objective $L_k^{(t)}$ is L-smooth: for all $\mathbf{U}, \mathbf{V} \in \mathbb{R}^d$,*

$$L_k^{(t)}(\mathbf{V}) \leq L_k^{(t)}(\mathbf{U}) + \big\langle \nabla L_k^{(t)}(\mathbf{U}), \mathbf{V} - \mathbf{U} \big\rangle + \frac{L}{2}\|\mathbf{V} - \mathbf{U}\|^2.$$

**Assumption 2** (Unbiased stochastic gradients). *For every task $t$, client $k$, global round $e$, and local step $s$, let $\xi_k^{(t,e,s)} \sim \mathcal{D}_k^{(t)}$. Then*

$$\mathbb{E}\left[\nabla_{\mathbf{W}}\ell\left(\mathbf{W}_k^{(t,e,s)}; \xi_k^{(t,e,s)}\right) \,\Big|\, \mathbf{W}_k^{(t,e,s)}\right] = \nabla L_k^{(t)}\left(\mathbf{W}_k^{(t,e,s)}\right).$$

**Assumption 3** (Bounded second moment of stochastic gradients). *There exists $G > 0$ such that for every task index $r \in \{1, \ldots, T\}$, every client $k \in [K]$, and every iterate $W$ visited by Algorithm 1,*

$$\mathbb{E}_{\xi \sim \mathcal{D}_k^{(r)}}\left\|\nabla_{\mathbf{W}}\ell(\mathbf{W}; \xi)\right\|^2 \le G^2.$$

**Assumption 4** (Current-task projected-gradient adequacy). *For each task $t$, there exists $\rho_t \in (0, 1]$ such that at every synchronization iterate $\mathbf{W}^{(t,e-1)}$, $e = 1, \ldots, E_t$,*

$$\left\|P^{(t-1)}\nabla L^{(t)}(\mathbf{W}^{(t,e-1)})\right\|^2 \ge \rho_t \left\|\nabla L^{(t)}(\mathbf{W}^{(t,e-1)})\right\|^2.$$

**Assumption 5** (Past-task interference coefficient). *For every pair of tasks $\tau < t$, there exists $\beta_\tau^{(t-1)} \in [0, 1]$ such that at every synchronization iterate $\mathbf{W}^{(t,e-1)}$ encountered while learning task $t$,*

$$\left\|P^{(t-1)}\nabla L^{(\tau)}(\mathbf{W}^{(t,e-1)})\right\| \le \beta_\tau^{(t-1)}\left\|\nabla L^{(\tau)}(\mathbf{W}^{(t,e-1)})\right\|.$$

**Remark 1** (Stability-plasticity assumption). *Assumption 4 measures* plasticity*: $\rho_t$ is the fraction of the current-task gradient energy that survives projection and therefore remains available for optimization. Assumption 5 measures* stability*: $\beta_\tau^{(t-1)}$ quantifies how much of an old-task gradient is still present inside the admissible update space while learning a later task. We do not identify $\rho_t$ or $\beta_\tau^{(t-1)}$ directly with the SVD energy threshold $\epsilon_l$. The threshold is defined in activation space, whereas $\rho_t$ and $\beta_\tau^{(t-1)}$ are gradient-space quantities. Relating them by a deterministic algebraic formula would require an additional representation–gradient alignment assumption, which we intentionally avoid.*

## A.3   Main results

We first state a task-wise convergence result for projected local training on the currently learned task. The theorem shows that, once the previously learned subspace is fixed, FedProTIP behaves like local SGD on the orthogonal complement of that subspace.

**Theorem 3** (Task-wise convergence on the current task). *Fix a task $t$ and abbreviate $\Delta_t \triangleq L^{(t)}(\mathbf{W}^{(t,0)}) - L^{(t)\star}$, where $L^{(t)\star} \triangleq \inf_{\mathbf{W}} L^{(t)}(\mathbf{W})$. Under Assumptions 1–4, the iterates generated while learning task $t$ satisfy*

$$\frac{1}{E_t}\sum_{e=1}^{E_t}\mathbb{E}\left\|\nabla L^{(t)}(\mathbf{W}^{(t,e-1)})\right\|^2 \le \frac{2\Delta_t}{\rho_t E_t S_t \eta_t} + \frac{L\eta_t S_t G^2}{\rho_t}\left(1 + \frac{L\eta_t S_t}{3\rho_t}\right). \tag{23}$$

The next theorem controls forgetting on earlier tasks. Since the empirical forgetting metric in Eq. (18) is defined in terms of accuracy, whereas the optimization analysis is naturally stated in terms of loss values, we work with the following loss-based analogue. For a past task $\tau < t$ and a future task $t$, define the round-$e$ loss increase

$$\Gamma_\tau^{(t,e)} \triangleq \mathbb{E}\left[L^{(\tau)}(\mathbf{W}^{(t,e)}) - L^{(\tau)}(\mathbf{W}^{(t,e-1)})\right]. \tag{24}$$

**Theorem 4** (Per-round and cumulative forgetting bounds). *Under Assumptions 1, 3, and 5, for every pair of tasks $\tau < t$ and every round $e = 1, \ldots, E_t$ during the training of task $t$,*

$$\Gamma_\tau^{(t,e)} \le \beta_\tau^{(t-1)} S_t \eta_t G^2 + \frac{L}{2}S_t^2 \eta_t^2 G^2. \tag{25}$$

*Consequently, the cumulative loss increase of task $\tau$ after all later tasks have been learned satisfies*

$$\mathbb{E}\left[L^{(\tau)}(\mathbf{W}^{(T)})\right] - \mathbb{E}\left[L^{(\tau)}(\mathbf{W}^{(\tau)})\right] \le G^2 \sum_{t=\tau+1}^{T} E_t\left(\beta_\tau^{(t-1)} S_t \eta_t + \frac{L}{2}S_t^2 \eta_t^2\right). \tag{26}$$

For later interpretation it is convenient to average the interference coefficients across all ordered pairs of past and future tasks:

$$\bar{\beta}_T \triangleq \frac{2}{T(T-1)} \sum_{t=2}^{T} \sum_{\tau=1}^{t-1} \beta_\tau^{(t-1)}. \tag{27}$$

We also define the loss-based average forgetting

$$\mathrm{FT}_{\mathrm{loss}}(T) \triangleq \frac{1}{T-1} \sum_{\tau=1}^{T-1} \left[ \mathbb{E}\Big[ L^{(\tau)}\big(\mathbf{W}^{(T)}\big) \Big] - \mathbb{E}\Big[ L^{(\tau)}\big(\mathbf{W}^{(\tau)}\big) \Big] \right]. \tag{28}$$

**Corollary 2** (Canonical step-size schedule). *Suppose the current-task step size for task t is chosen as*

$$\eta_t = \frac{1}{L S_t \sqrt{E_t}}. \tag{29}$$

*Then Theorem 3 yields*

$$\frac{1}{E_t} \sum_{e=1}^{E_t} \mathbb{E}\Big\| \nabla L^{(t)}\big(\mathbf{W}^{(t,e-1)}\big) \Big\|^2 \le \frac{2L\Delta_t}{\rho_t \sqrt{E_t}} + \frac{G^2}{\rho_t \sqrt{E_t}} \left( 1 + \frac{1}{3\rho_t \sqrt{E_t}} \right), \tag{30}$$

*so the current-task stationarity measure decays as $\mathcal{O}(E_t^{-1/2})$. Moreover, Eq. (26) becomes*

$$\mathbb{E}\Big[ L^{(\tau)}\big(\mathbf{W}^{(T)}\big) \Big] - \mathbb{E}\Big[ L^{(\tau)}\big(\mathbf{W}^{(\tau)}\big) \Big] \le \frac{G^2}{L} \sum_{t=\tau+1}^{T} \left( \beta_\tau^{(t-1)} \sqrt{E_t} + \frac{1}{2} \right). \tag{31}$$

*If the hyperparameters are common across tasks, i.e. $E_t \equiv E$, $S_t \equiv S$, and $\eta_t \equiv \eta$, then*

$$\mathrm{FT}_{\mathrm{loss}}(T) \le \frac{T G^2 \bar{\beta}_T}{2L} \sqrt{E} + \frac{T G^2}{4L}. \tag{32}$$

## B  Proofs for the convergence analysis

### B.1  Auxiliary lemmas

For a fixed task $t$, we abbreviate

$$L \equiv L^{(t)}, \qquad P \equiv P^{(t-1)}, \qquad E \equiv E_t, \qquad S \equiv S_t, \qquad \eta \equiv \eta_t, \qquad \rho \equiv \rho_t,$$

and write $\mathbf{W}^{(e)} \equiv \mathbf{W}^{(t,e)}$ and $\mathbf{W}_k^{(e,s)} \equiv \mathbf{W}_k^{(t,e,s)}$ whenever there is no ambiguity.

**Lemma 1** (Projection is non-expansive). *The matrix $P^{(t-1)}$ is symmetric and idempotent, with $\|P^{(t-1)}\|_2 = 1$. Consequently, for every $v \in \mathbb{R}^d$,*

$$\big\| P^{(t-1)} v \big\| \le \|v\|.$$

*Moreover, Assumptions 2 and 3 are preserved after projection:*

$$\mathbb{E}\big[ \widetilde{\mathbf{g}}_k^{(t,e,s)} \mid \mathbf{W}_k^{(t,e,s)} \big] = P^{(t-1)} \nabla L_k^{(t)}\big( \mathbf{W}_k^{(t,e,s)} \big),$$

$$\mathbb{E}\big\| \widetilde{\mathbf{g}}_k^{(t,e,s)} \big\|^2 \le G^2.$$

*Proof.* Since $\mathbf{\Phi}^{(1:t-1)}$ has orthonormal columns, $\big(\mathbf{\Phi}^{(1:t-1)}\big)^\top \mathbf{\Phi}^{(1:t-1)} = I$. Therefore

$$\big( P^{(t-1)} \big)^2 = \big( I - \mathbf{\Phi}^{(1:t-1)}(\mathbf{\Phi}^{(1:t-1)})^\top \big)^2 = I - \mathbf{\Phi}^{(1:t-1)}(\mathbf{\Phi}^{(1:t-1)})^\top = P^{(t-1)}.$$

Symmetry is immediate, so $P^{(t-1)}$ is an orthogonal projector. Its eigenvalues lie in $\{0, 1\}$, hence $\|P^{(t-1)}\|_2 = 1$ and $\|P^{(t-1)} v\| \le \|v\|$ for all $v$. The unbiasedness statement follows from the linearity of $P^{(t-1)}$ and Assumption 2. The second-moment bound follows from non-expansiveness and Assumption 3. $\qquad\square$

**Lemma 2** (Orthogonal confinement of each round update). *For every task $t$, round $e$, and client $k$,*

$$\left(\mathbf{\Phi}^{(1:t-1)}\right)^{\top}\left(\mathbf{W}_k^{(t,e,S_t)} - \mathbf{W}^{(t,e-1)}\right) = \mathbf{0}.$$

*Consequently, $\mathbf{W}^{(t,e)} - \mathbf{W}^{(t,e-1)} \in \text{range}(P^{(t-1)})$ for every round $e$.*

*Proof.* Telescoping the local updates gives

$$\mathbf{W}_k^{(t,e,S_t)} - \mathbf{W}^{(t,e-1)} = -\eta_t \sum_{s=0}^{S_t-1} P^{(t-1)}\mathbf{g}_k^{(t,e,s)}.$$

Left-multiplying by $\left(\mathbf{\Phi}^{(1:t-1)}\right)^{\top}$ and using $\left(\mathbf{\Phi}^{(1:t-1)}\right)^{\top}P^{(t-1)} = 0$ proves the first claim. Averaging over clients with weights $p_k^{(t)}$ proves the second. $\qquad\square$

**Lemma 3** (Local drift bound). *Under Assumption 3, for every client $k$ and every local step $s \in \{0,\ldots,S_t\}$,*

$$\mathbb{E}\left\|\mathbf{W}_k^{(t,e,s)} - \mathbf{W}^{(t,e-1)}\right\|^2 \le s^2\eta_t^2 G^2.$$

*Proof.* Using the telescoping representation,

$$\mathbf{W}_k^{(t,e,s)} - \mathbf{W}^{(t,e-1)} = -\eta_t \sum_{j=0}^{s-1} P^{(t-1)}\mathbf{g}_k^{(t,e,j)}.$$

By Jensen's inequality and Lemma 1,

$$\mathbb{E}\left\|\mathbf{W}_k^{(t,e,s)} - \mathbf{W}^{(t,e-1)}\right\|^2 \le s\eta_t^2 \sum_{j=0}^{s-1} \mathbb{E}\left\|P^{(t-1)}\mathbf{g}_k^{(t,e,j)}\right\|^2$$

$$\le s\eta_t^2 \sum_{j=0}^{s-1} \mathbb{E}\left\|\mathbf{g}_k^{(t,e,j)}\right\|^2$$

$$\le s^2\eta_t^2 G^2.$$

$\square$

## B.2 Proof of Theorem 3

*Proof of Theorem 3.* Fix a task $t$ and suppress the task index as described above. Let

$$\Delta^{(e)} \triangleq \mathbf{W}^{(e)} - \mathbf{W}^{(e-1)}.$$

By $L$-smoothness of $L$, for each round $e$,

$$L\left(\mathbf{W}^{(e)}\right) \le L\left(\mathbf{W}^{(e-1)}\right) + \left\langle \nabla L\left(\mathbf{W}^{(e-1)}\right), \Delta^{(e)}\right\rangle + \frac{L}{2}\left\|\Delta^{(e)}\right\|^2. \tag{33}$$

We bound the two extra terms separately.

**Step 1: the first-order term.** Using Eq. (22),

$$\Delta^{(e)} = -\eta \sum_{s=0}^{S-1} \sum_{k=1}^{K} p_k^{(t)} \widetilde{\mathbf{g}}_k^{(t,e,s)}.$$

Taking conditional expectation and applying Lemma 1 gives

$$
\mathbb{E}\Big[\big\langle \nabla L(\mathbf{W}^{(e-1)}),\, \Delta^{(e)}\big\rangle\Big]
$$
$$
= -\eta \sum_{s=0}^{S-1} \mathbb{E}\left\langle \nabla L(\mathbf{W}^{(e-1)}),\, P \sum_{k=1}^{K} p_k^{(t)} \nabla L_k\big(\mathbf{W}_k^{(e,s)}\big)\right\rangle.
\tag{34}
$$

Define the local-drift error

$$
\mathbf{V}^{(e,s)} \triangleq \sum_{k=1}^{K} p_k^{(t)} \Big[\nabla L_k\big(\mathbf{W}_k^{(e,s)}\big) - \nabla L_k\big(\mathbf{W}^{(e-1)}\big)\Big].
$$

Since $\sum_k p_k^{(t)} \nabla L_k(\mathbf{W}^{(e-1)}) = \nabla L(\mathbf{W}^{(e-1)})$, Eq. (34) becomes

$$
\mathbb{E}\Big[\big\langle \nabla L(\mathbf{W}^{(e-1)}),\, \Delta^{(e)}\big\rangle\Big]
$$
$$
= -\eta S\, \mathbb{E}\big\|P\nabla L\big(\mathbf{W}^{(e-1)}\big)\big\|^2 - \eta \sum_{s=0}^{S-1} \mathbb{E}\big\langle \nabla L(\mathbf{W}^{(e-1)}),\, P\mathbf{V}^{(e,s)}\big\rangle.
\tag{35}
$$

By Assumption 4,

$$
\big\|P\nabla L\big(\mathbf{W}^{(e-1)}\big)\big\|^2 \geq \rho \big\|\nabla L\big(\mathbf{W}^{(e-1)}\big)\big\|^2.
$$

Next, by Jensen's inequality, $L$-smoothness, and Lemma 3,

$$
\mathbb{E}\big\|\mathbf{V}^{(e,s)}\big\|^2 \leq \sum_{k=1}^{K} p_k^{(t)} \mathbb{E}\big\|\nabla L_k\big(\mathbf{W}_k^{(e,s)}\big) - \nabla L_k\big(\mathbf{W}^{(e-1)}\big)\big\|^2
$$
$$
\leq L^2 \sum_{k=1}^{K} p_k^{(t)} \mathbb{E}\big\|\mathbf{W}_k^{(e,s)} - \mathbf{W}^{(e-1)}\big\|^2
$$
$$
\leq L^2 s^2 \eta^2 G^2.
$$

Therefore, Young's inequality implies

$$
\big|\big\langle \nabla L\big(\mathbf{W}^{(e-1)}\big),\, P\mathbf{V}^{(e,s)}\big\rangle\big| \leq \frac{\rho}{2}\big\|\nabla L\big(\mathbf{W}^{(e-1)}\big)\big\|^2 + \frac{1}{2\rho}\big\|\mathbf{V}^{(e,s)}\big\|^2.
$$

Substituting these estimates into Eq. (35) and using $\sum_{s=0}^{S-1} s^2 \leq S^3/3$ gives

$$
\mathbb{E}\Big[\big\langle \nabla L(\mathbf{W}^{(e-1)}),\, \Delta^{(e)}\big\rangle\Big] \leq -\frac{\rho\eta S}{2}\mathbb{E}\big\|\nabla L\big(\mathbf{W}^{(e-1)}\big)\big\|^2 + \frac{L^2\eta^3 S^3 G^2}{6\rho}.
\tag{36}
$$

**Step 2: the quadratic term.** By Eq. (22), Jensen's inequality, and Lemma 1,

$$
\mathbb{E}\big\|\Delta^{(e)}\big\|^2 = \eta^2 \mathbb{E}\left\|\sum_{s=0}^{S-1}\sum_{k=1}^{K} p_k^{(t)} \widetilde{\mathbf{g}}_k^{(t,e,s)}\right\|^2
$$
$$
\leq \eta^2 S \sum_{s=0}^{S-1} \mathbb{E}\left\|\sum_{k=1}^{K} p_k^{(t)} \widetilde{\mathbf{g}}_k^{(t,e,s)}\right\|^2
$$
$$
\leq \eta^2 S \sum_{s=0}^{S-1}\sum_{k=1}^{K} p_k^{(t)} \mathbb{E}\big\|\widetilde{\mathbf{g}}_k^{(t,e,s)}\big\|^2
$$
$$
\leq \eta^2 S^2 G^2.
\tag{37}
$$

Hence,

$$
\frac{L}{2}\mathbb{E}\big\|\Delta^{(e)}\big\|^2 \leq \frac{L}{2}\eta^2 S^2 G^2.
\tag{38}
$$

**Step 3: combine and telescope.** Taking expectation in Eq. (33) and substituting Eqs. (36) and (38) gives

$$\mathbb{E}L\big(\mathbf{W}^{(e)}\big) \leq \mathbb{E}L\big(\mathbf{W}^{(e-1)}\big) - \frac{\rho\eta S}{2}\mathbb{E}\big\|\nabla L\big(\mathbf{W}^{(e-1)}\big)\big\|^2$$
$$+ \frac{L^2\eta^3 S^3 G^2}{6\rho} + \frac{L}{2}\eta^2 S^2 G^2.$$

Rearranging, summing over $e = 1, \ldots, E$, and using $L(\mathbf{W}^{(E)}) \geq L^\star$ yields

$$\frac{1}{E}\sum_{e=1}^{E}\mathbb{E}\big\|\nabla L\big(\mathbf{W}^{(e-1)}\big)\big\|^2 \leq \frac{2\big(L(\mathbf{W}^{(0)}) - L^\star\big)}{\rho E S \eta} + \frac{L\eta S G^2}{\rho} + \frac{L^2\eta^2 S^2 G^2}{3\rho^2},$$

which is exactly Eq. (23). $\qquad\qquad\square$

### B.3 Proof of Theorem 4

*Proof of Theorem 4.* Fix two tasks $\tau < t$. For round $e$ of task $t$, define

$$\Delta^{(t,e)} \triangleq \mathbf{W}^{(t,e)} - \mathbf{W}^{(t,e-1)}.$$

By Lemma 2, $\Delta^{(t,e)} \in \text{range}(P^{(t-1)})$. Applying $L$-smoothness to the old-task loss $L^{(\tau)}$ at the synchronization iterate $\mathbf{W}^{(t,e-1)}$ gives

$$L^{(\tau)}\big(\mathbf{W}^{(t,e)}\big) \leq L^{(\tau)}\big(\mathbf{W}^{(t,e-1)}\big) + \big\langle\nabla L^{(\tau)}\big(\mathbf{W}^{(t,e-1)}\big), \Delta^{(t,e)}\big\rangle + \frac{L}{2}\big\|\Delta^{(t,e)}\big\|^2. \qquad (39)$$

Because $\Delta^{(t,e)} \in \text{range}(P^{(t-1)})$,

$$\big\langle\nabla L^{(\tau)}\big(\mathbf{W}^{(t,e-1)}\big), \Delta^{(t,e)}\big\rangle = \big\langle P^{(t-1)}\nabla L^{(\tau)}\big(\mathbf{W}^{(t,e-1)}\big), \Delta^{(t,e)}\big\rangle.$$

Hence, by Assumption 5,

$$\big\langle\nabla L^{(\tau)}\big(\mathbf{W}^{(t,e-1)}\big), \Delta^{(t,e)}\big\rangle \leq \beta_\tau^{(t-1)}\big\|\nabla L^{(\tau)}\big(\mathbf{W}^{(t,e-1)}\big)\big\| \cdot \big\|\Delta^{(t,e)}\big\|. \qquad (40)$$

Now let $\mathbf{W} = \mathbf{W}^{(t,e-1)}$ be a synchronization iterate encountered while learning task $t$. By Assumption 3, for every client $k$,

$$\big\|\nabla L_k^{(\tau)}(\mathbf{W})\big\|^2 = \big\|\mathbb{E}_{\xi\sim\mathcal{D}_k^{(\tau)}}[\nabla_{\mathbf{w}}\ell(\mathbf{W};\xi)]\big\|^2 \leq \mathbb{E}_{\xi\sim\mathcal{D}_k^{(\tau)}}\|\nabla_{\mathbf{w}}\ell(\mathbf{W};\xi)\|^2 \leq G^2.$$

by Jensen's inequality. Therefore,

$$\|\nabla L^{(\tau)}(\mathbf{W})\| = \left\|\sum_{k=1}^{K}p_k^{(\tau)}\nabla L_k^{(\tau)}(\mathbf{W})\right\| \leq \sum_{k=1}^{K}p_k^{(\tau)}\|\nabla L_k^{(\tau)}(\mathbf{W})\| \leq G.$$

Next,

$$\mathbb{E}\big\|\Delta^{(t,e)}\big\| = \mathbb{E}\left\|\eta_t\sum_{s=0}^{S_t-1}\sum_{k=1}^{K}p_k^{(t)}P^{(t-1)}\mathbf{g}_k^{(t,e,s)}\right\|$$
$$\leq \eta_t\sum_{s=0}^{S_t-1}\sum_{k=1}^{K}p_k^{(t)}\mathbb{E}\big\|P^{(t-1)}\mathbf{g}_k^{(t,e,s)}\big\|$$
$$\leq \eta_t S_t G,$$

and similarly,

$$\mathbb{E}\big\|\Delta^{(t,e)}\big\|^2 \leq \eta_t^2 S_t^2 G^2.$$

Taking expectation in Eq. (39) and using the bounds above yields

$$\Gamma_\tau^{(t,e)} \leq \beta_\tau^{(t-1)} S_t \eta_t G^2 + \frac{L}{2} S_t^2 \eta_t^2 G^2,$$

which proves Eq. (25).

Summing Eq. (25) over rounds $e = 1, \ldots, E_t$ gives the loss increase of task $\tau$ while training task $t$. Summing the resulting inequality over all future tasks $t = \tau + 1, \ldots, T$ yields Eq. (26). $\qquad\square$

### B.4 Proof of Corollary 2

*Proof of Corollary 2.* Substituting $\eta_t = 1/(LS_t\sqrt{E_t})$ into Eq. (23) gives

$$\frac{1}{E_t} \sum_{e=1}^{E_t} \mathbb{E}\big\|\nabla L^{(t)}\big(\mathbf{W}^{(t,e-1)}\big)\big\|^2 \leq \frac{2L\Delta_t}{\rho_t\sqrt{E_t}} + \frac{G^2}{\rho_t\sqrt{E_t}}\left(1 + \frac{1}{3\rho_t\sqrt{E_t}}\right),$$

which is Eq. (30).

Next, substituting the same step size into Eq. (26) gives

$$\mathbb{E}\Big[L^{(\tau)}\big(\mathbf{W}^{(T)}\big)\Big] - \mathbb{E}\Big[L^{(\tau)}\big(\mathbf{W}^{(\tau)}\big)\Big] \leq G^2 \sum_{t=\tau+1}^{T} E_t \left(\frac{\beta_\tau^{(t-1)}}{L\sqrt{E_t}} + \frac{1}{2LE_t}\right)$$

$$= \frac{G^2}{L} \sum_{t=\tau+1}^{T} \left(\beta_\tau^{(t-1)}\sqrt{E_t} + \frac{1}{2}\right),$$

which is Eq. (31).

Finally, under common hyperparameters,

$$\mathrm{FT}_{\mathrm{loss}}(T) \leq \frac{G^2}{T-1} \sum_{\tau=1}^{T-1} \sum_{t=\tau+1}^{T} E\left(\beta_\tau^{(t-1)} S\eta + \frac{L}{2} S^2 \eta^2\right)$$

$$= \frac{G^2 ES\eta}{T-1} \sum_{t=2}^{T} \sum_{\tau=1}^{t-1} \beta_\tau^{(t-1)} + \frac{LG^2 ES^2\eta^2}{2(T-1)} \sum_{\tau=1}^{T-1} (T-\tau)$$

$$= \frac{TG^2 \bar{\beta}_T}{2} ES\eta + \frac{TLG^2}{4} ES^2\eta^2.$$

Substituting $\eta = 1/(LS\sqrt{E})$ gives Eq. (32). $\qquad\square$

## C  Correctness of global subspace aggregation

The previous results analyze the optimization dynamics once the subspace $\mathbf{\Phi}^{(1:t-1)}$ has already been formed. We conclude by recording a simple structural fact about the server aggregation rule of Section 4.3: it preserves the union of the client-transmitted task subspaces.

**Proposition 1** (Global aggregation spans the union of local bases). *Fix a layer $l$ and a task $t$. Let the client-side bases extracted after task $t$ be $U_{l,1}^{(t)}, \ldots, U_{l,K}^{(t)}$, each with orthonormal columns. If the server forms $\Phi_l^{(t)}$ by the iterative orthogonal appending rule described in Section 4.3, then*

$$\mathrm{span}\big(\Phi_l^{(t)}\big) = \mathrm{span}\Big(\bigcup_{k=1}^{K} U_{l,k}^{(t)}\Big).$$

*Proof.* Let $\Phi_{l,[k]}^{(t)}$ denote the intermediate aggregated basis after incorporating the first $k$ clients. We prove by induction on $k$ that

$$\mathrm{span}\big(\Phi_{l,[k]}^{(t)}\big) = \mathrm{span}\Big(\bigcup_{j=1}^{k} U_{l,j}^{(t)}\Big).$$

The statement is immediate for $k = 1$ because the algorithm initializes $\Phi_{l,[1]}^{(t)} = U_{l,1}^{(t)}$.

Assume the claim holds for $k-1$. At step $k$, the server computes the residual

$$R_{l,k}^{(t)} = U_{l,k}^{(t)} - \Phi_{l,[k-1]}^{(t)} \big(\Phi_{l,[k-1]}^{(t)}\big)^{\top} U_{l,k}^{(t)},$$

which is exactly the component of $U_{l,k}^{(t)}$ orthogonal to the current aggregated subspace. Appending an orthonormal basis of $R_{l,k}^{(t)}$ therefore adds precisely the new directions in $U_{l,k}^{(t)}$ that were not already present in $\mathrm{span}(\Phi_{l,[k-1]}^{(t)})$. Hence,

$$\mathrm{span}\big(\Phi_{l,[k]}^{(t)}\big) = \mathrm{span}\big(\Phi_{l,[k-1]}^{(t)}\big) + \mathrm{span}\big(U_{l,k}^{(t)}\big) = \mathrm{span}\Big(\bigcup_{j=1}^{k} U_{l,j}^{(t)}\Big),$$

which proves the induction step. Taking $k = K$ completes the proof. $\square$

## D  Additional Experimental Results

### D.1  Results on Different Models

To assess whether FedProTIP's gains depend on a particular backbone or pretraining regime, we additionally evaluate (i) a ResNet18 trained from scratch and (ii) a pretrained ViT-B/16 on 10-split CIFAR100.

We report the results using a scratch-trained ResNet18 in Table 6. In the task-agnostic inference setting, our method achieves the best performance, showing a significant margin over all other baselines. We set the task identity prediction threshold to $\epsilon_l = 0.95$, $\forall l$, based on a hyperparameter search. This threshold is higher than the one used for the pretrained ResNet18 model ($\epsilon_l = 0.7$), as the pretrained model provides a stronger feature extractor that better generalizes across tasks. In contrast, when training from scratch, preserving knowledge of previous tasks becomes more critical, hence the need for a higher threshold. As shown in Table 6, while FedProTIP is not the best-performing method in the task-aware inference scenario, where the ground-truth task ID is available during testing, it outperforms all baselines in the more practical task-agnostic setting with a large margin. Our method achieves the highest accuracy and lowest forgetting, primarily due to effective task identity prediction.

We also evaluate our method on a different backbone, pre-trained ViT-B/16 (Steiner et al., 2021), as reported in Table 7. In this setting, CIFAR100 images ($3 \times 32 \times 32$) are resized to $224 \times 224$ to match the ViT input resolution. We set the number of local epochs to 5 and perform 20 global rounds per task. For TARGET and LANDER, we follow the stronger protocol of generating synthetic images at the native $32 \times 32$ resolution and then applying resizing augmentation, which improves their performance in this setting. FedProTIP attains near-ceiling task-agnostic accuracy (98.38% ACC) with negligible forgetting (0.20% FT), suggesting that the learned subspaces remain highly stable and that task routing is reliable with transformer features. In addition, FedProTIP without TIP achieves competitive task-agnostic accuracy while substantially reducing forgetting, indicating that the projection mechanism transfers effectively across architectures.

### D.2  Impact of Sampling Dimension

We perform an ablation study on the target feature dimension $m_s$ used in randomized SVD for extracting core bases in Table 8. We vary $m_s \in \{2048, 1024, 512\}$ while keeping all other settings fixed, and report accuracy, forgetting, and runtime averaged over three seeds. Across all three choices, performance is stable. Both task-aware and task-agnostic accuracy vary within 1.4%, and TIP consistently improves task-agnostic

Table 6: Metrics (%) of accuracy (↑) and forgetting (↓) on 10-split CIFAR100 using **ResNet18 from scratch**. We report the average accuracy and standard deviation over 3 trials, each with different seeds. $\epsilon_l = 0.95$ is used for FedProTIP.

| Method | Task-Agnostic | | Task-Aware | |
|---|---|---|---|---|
| | ACC | FT | ACC | FT |
| FedAvg | $11.04_{\pm 0.37}$ | $54.90_{\pm 1.62}$ | $36.87_{\pm 1.36}$ | $42.69_{\pm 1.00}$ |
| Target | $23.05_{\pm 1.93}$ | $\underline{9.00}_{\pm 1.15}$ | $\underline{71.86}_{\pm 0.55}$ | $2.32_{\pm 0.66}$ |
| Lander | $\underline{29.37}_{\pm 1.09}$ | $20.17_{\pm 1.90}$ | $\mathbf{73.69}_{\pm 0.64}$ | $1.39_{\pm 0.38}$ |
| FOT | $22.18_{\pm 1.35}$ | $9.10_{\pm 0.57}$ | $67.07_{\pm 1.36}$ | $\underline{0.73}_{\pm 0.06}$ |
| FedProTIP (-t) | $24.84_{\pm 0.91}$ | $12.29_{\pm 1.54}$ | $68.75_{\pm 1.71}$ | $\mathbf{0.68}_{\pm 0.48}$ |
| FedProTIP | $\mathbf{65.77}_{\pm 1.92}$ | $\mathbf{2.38}_{\pm 0.75}$ | — | — |

Table 7: Metrics (%) of accuracy (↑) and forgetting (↓) on 10-split CIFAR100 using **ViT-B/16**. We report the average and standard deviation over 2 trials with different seeds. $\epsilon_l = 0.7$ is used for FedProTIP.

| Method | Task-Agnostic | | Task-Aware | |
|---|---|---|---|---|
| | ACC | FT | ACC | FT |
| FedAvg | $67.15_{\pm 4.40}$ | $22.34_{\pm 0.98}$ | $95.18_{\pm 0.69}$ | $2.88_{\pm 0.33}$ |
| Target | $\underline{81.50}_{\pm 0.06}$ | $7.33_{\pm 1.34}$ | $\underline{98.30}_{\pm 0.12}$ | $\underline{0.37}_{\pm 0.01}$ |
| Lander | $61.43_{\pm 5.53}$ | $27.33_{\pm 4.31}$ | $96.07_{\pm 1.05}$ | $2.39_{\pm 0.97}$ |
| FOT | $72.27_{\pm 0.79}$ | $21.73_{\pm 0.46}$ | $96.46_{\pm 0.20}$ | $2.28_{\pm 0.15}$ |
| FedProTIP (-t) | $79.90_{\pm 1.46}$ | $\underline{4.54}_{\pm 0.35}$ | $\mathbf{98.36}_{\pm 0.04}$ | $\mathbf{0.15}_{\pm 0.09}$ |
| FedProTIP | $\mathbf{98.38}_{\pm 0.02}$ | $\mathbf{0.20}_{\pm 0.08}$ | — | — |

inference. Interestingly, the best overall performance is obtained at $m_s = 512$, with slightly lower accuracy at 1024 and 2048. This trend suggests that larger dimensions may retain redundant directions that capture client-specific noise rather than task-relevant structure. Protecting these directions enlarges the preserved subspace and can overly constrain subsequent learning, reducing forward transfer. In terms of efficiency, runtime increases markedly as $m_s$ grows, with both base extraction and total training time rising substantially. Since accuracy remains largely stable across values of $m_s$, we use $m_s = 512$ as the default because it offers the best balance between efficiency and performance.

### D.3 Batch Size Sensitivity

We evaluate FedProTIP and baselines with batch sizes 32, 64, and 128 (Table 10) under both task-aware and task-agnostic inference. Across all settings, FedProTIP consistently outperforms prior methods. The relative gain from TIP is smaller at low batch sizes, since limited samples increase the variance of final-layer activations, injecting noise into the relevance vector and cosine similarities. As batch size grows, variance decreases, stabilizing TIP and amplifying its benefits. Even in the small-batch regime, however, FedProTIP still yields meaningful improvements under task-agnostic inference.

### D.4 Different Threshold Values in FedProTIP

We present the results of FedProTIP with different threshold values on CIFAR100, DomainNet, and ImageNet-R in Table 11, evaluating thresholds of 0.7, 0.8, and 0.9 in terms of both average accuracy and forgetting. Across all three datasets, FedProTIP maintains stable accuracy in both task-aware and task-agnostic settings, showing only minor sensitivity to the choice of threshold.

On CIFAR100, forgetting in the task-agnostic case remains positive but steadily decreases as the threshold increases, while on ImageNet-R, a similar trend is observed, culminating in negative forgetting at $\epsilon_l = 0.9$. Negative forgetting arises because the task-identity predictor improves as more tasks are introduced, retroactively correcting earlier misclassifications. At early stages, the predictor is poorly calibrated and

Table 8: Ablation on the target feature dimension $d$ used in randomized SVD. We report the metrics under task-aware, task-agnostic, and with TIP settings.

| $m_s$ | ACC ($\uparrow$) | | | FT ($\downarrow$) | | | Runtime (h) | |
|---|---|---|---|---|---|---|---|---|
| | Aware | Agnostic | + TIP | Aware | Agnostic | + TIP | Base | Total |
| 2048 | 85.53 | 46.07 | 84.32 | 1.41 | 16.62 | 0.63 | 2.26 | 3.42 |
| 1024 | 86.03 | 47.04 | 85.68 | 1.75 | 16.55 | 0.10 | 1.45 | 2.62 |
| 512 | 86.26 | 48.41 | 86.00 | 0.96 | 15.59 | 0.83 | 1.25 | 2.35 |

Table 9: Per-task and per-client communication cost (MB) comparison between FOT (Model+Act) and FedProTIP (Model+Base+Ref) in 10-split CIFAR100.

| Task | 1 | 2 | 3 | 4 | 5 | 6 | 7 | 8 | 9 | 10 |
|---|---|---|---|---|---|---|---|---|---|---|
| Model | 46.686 | 46.764 | 46.842 | 46.920 | 46.998 | 47.077 | 47.155 | 47.233 | 47.311 | 47.389 |
| Act | 48 | 48 | 48 | 48 | 48 | 48 | 48 | 48 | 48 | 48 |
| Base | 9.794 | 5.646 | 2.841 | 1.990 | 1.278 | 0.693 | 0.489 | 0.434 | 0.442 | 0.264 |
| Ref | 0.000 | 0.000 | 0.000 | 0.000 | 0.000 | 0.000 | 0.000 | 0.000 | 0.000 | 0.001 |

Table 10: Metrics computed in the experiments on 10-Split CIFAR100 with $\alpha = 0.5$ and varying batch sizes $\{32, 64, 128\}$.

| Batch Size | Method | Task-Aware | Task-Agnostic | +TIP |
|---|---|---|---|---|
| 32 | FedAvg | 24.82 | 10.74 | - |
| | GLFC | 81.56 | 13.28 | - |
| | LGA | 82.59 | 13.03 | - |
| | TARGET | 74.29 | 31.67 | - |
| | LANDER | 78.33 | 32.30 | - |
| | FOT | 83.38 | 41.05 | - |
| | FedProTIP | **87.86** | **48.40** | **84.22** |
| 64 | FedAvg | 38.99 | 15.35 | - |
| | GLFC | 75.26 | 22.86 | - |
| | LGA | 85.04 | 14.35 | - |
| | TARGET | 69.81 | 27.37 | - |
| | LANDER | 84.19 | 37.59 | - |
| | FOT | 82.59 | 41.80 | - |
| | FedProTIP | **86.26** | **48.41** | **86.00** |
| 128 | FedAvg | 53.65 | 20.67 | - |
| | GLFC | 73.25 | 12.40 | - |
| | LGA | 75.25 | 11.53 | - |
| | TARGET | 67.58 | 26.70 | - |
| | LANDER | 79.83 | 30.30 | - |
| | FOT | 81.61 | 39.94 | - |
| | FedProTIP | **85.20** | **45.80** | **85.20** |

often misassigns samples from earlier tasks, but later tasks provide richer contrast and sharpen decision boundaries, boosting measured accuracy on prior tasks. The threshold parameter $\epsilon_l$ also plays a critical role. A higher threshold enforces stricter preservation of gradient subspaces, biasing the stability–plasticity trade-off toward stability. In practice, this means that representations associated with earlier tasks are less likely to be overwritten when new tasks arrive. As a result, catastrophic forgetting is reduced, and in some cases (e.g., ImageNet-R at $\epsilon_l = 0.9$) the combination of preserved subspaces and improved task-identity prediction even yields negative forgetting.

Finally, across all datasets and thresholds, the with TIP setting shows minimal sensitivity to threshold choice in terms of accuracy. However, excessively high thresholds can overemphasize stability, limiting plasticity and thereby reducing the learnability of new tasks.

## D.5 Different Task Orders

We present results for different task orderings in DomainNet. Table 12, which is also presented in the main paper, the task order is as follows: (clipart → real → painting → sketch → infograph → quickdraw). Recognizing that DomainNet exhibits varying levels of task/domain similarity, we include Table 13 to report results under a second ordering: (clipart → infograph → painting → quickdraw → real → sketch). These results show that FedProTIP consistently achieves strong performance regardless of task order, highlighting its robustness to domain heterogeneity and variations in task scheduling. This trend holds across both orderings,

Table 11: Metrics computed from FedProTIP experiments ($\alpha = 0.5$) with different thresholds $\epsilon_l$.

| Dataset | Threshold | Task-Aware | | Task-Agnostic | | +TIP | |
|---|---|---|---|---|---|---|---|
| | | ACC | FT | ACC | FT | ACC | FT |
| 10-split CIFAR100 | 0.7 | 86.26 | 1.23 | 48.41 | 15.59 | 86.00 | 1.26 |
| | 0.8 | 86.46 | 0.38 | 49.04 | 12.90 | 85.40 | 1.35 |
| | 0.9 | 85.88 | 0.08 | 47.91 | 11.80 | 85.59 | 0.15 |
| 10-split ImageNet-R | 0.7 | 61.72 | 2.35 | 35.64 | 8.65 | 54.48 | 7.48 |
| | 0.8 | 61.37 | 3.07 | 37.18 | 8.07 | 59.19 | 1.38 |
| | 0.9 | 58.99 | 1.63 | 33.80 | 6.35 | 53.41 | -2.16 |
| 6-split DomainNet | 0.7 | 28.75 | 1.45 | 28.85 | 6.43 | 25.30 | 3.76 |
| | 0.8 | 29.20 | 1.06 | 29.21 | 5.36 | 27.97 | 1.53 |
| | 0.9 | 28.99 | 0.72 | 27.35 | 6.41 | 27.78 | 0.76 |

with FedProTIP without TIP providing greater advantages in domain-incremental learning. Adapting domain-incremental specific modules in conjunction with TIP represents a promising direction for future research.

Table 12: Metrics (%) of accuracy ($\uparrow$) and forgetting ($\downarrow$) computed in the experiments on 6-split DomainNet of order (clipart $\rightarrow$ real $\rightarrow$ painting $\rightarrow$ sketch $\rightarrow$ infograph $\rightarrow$ quickdraw). We report the average accuracy and standard deviation over 2 trials, each with different seeds.

| Method | 6-Split DomainNet | | | | | |
|---|---|---|---|---|---|---|
| | IID | | $\alpha = 0.5$ | | $\alpha = 0.2$ | |
| | ACC | FT | ACC | FT | ACC | FT |
| FedAvg | $10.79_{\pm 0.18}$ | $27.74_{\pm 0.83}$ | $10.72_{\pm 0.14}$ | $25.66_{\pm 0.47}$ | $10.53_{\pm 0.20}$ | $25.57_{\pm 0.27}$ |
| TARGET | $21.53_{\pm 0.93}$ | $9.73_{\pm 0.51}$ | $20.61_{\pm 0.18}$ | $7.89_{\pm 0.54}$ | $20.64_{\pm 0.90}$ | $8.31_{\pm 1.12}$ |
| LANDER | $21.88_{\pm 0.32}$ | $8.90_{\pm 0.13}$ | $21.59_{\pm 1.00}$ | $10.27_{\pm 0.75}$ | $22.11_{\pm 0.26}$ | $8.59_{\pm 0.85}$ |
| FOT | $24.59_{\pm 1.00}$ | $8.85_{\pm 0.30}$ | $24.13_{\pm 0.25}$ | $8.44_{\pm 0.31}$ | $23.84_{\pm 0.00}$ | $8.33_{\pm 0.15}$ |
| FedProTIP (-t) | $\mathbf{29.64}_{\pm 0.86}$ | $\underline{6.38}_{\pm 0.09}$ | $\mathbf{28.85}_{\pm 1.46}$ | $\underline{6.43}_{\pm 0.54}$ | $\mathbf{28.74}_{\pm 0.17}$ | $\underline{6.14}_{\pm 0.33}$ |
| **FedProTIP** | $\underline{27.60}_{\pm 0.91}$ | $\mathbf{2.89}_{\pm 0.43}$ | $\underline{25.30}_{\pm 0.30}$ | $\mathbf{3.76}_{\pm 1.14}$ | $\underline{25.98}_{\pm 0.18}$ | $\mathbf{2.88}_{\pm 0.01}$ |

Table 13: Metrics (%) of accuracy ($\uparrow$) and forgetting ($\downarrow$) computed in the experiments on 6-split DomainNet of order (clipart $\rightarrow$ infograph $\rightarrow$ painting $\rightarrow$ quickdraw $\rightarrow$ real $\rightarrow$ sketch). We report the average accuracy and standard deviation over 2 trials, each with different seeds.

| Method | 6-Split DomainNet | | | | | |
|---|---|---|---|---|---|---|
| | IID | | $\alpha = 0.5$ | | $\alpha = 0.2$ | |
| | ACC | FT | ACC | FT | ACC | FT |
| FedAvg | $20.34_{\pm 0.74}$ | $17.16_{\pm 0.33}$ | $20.53_{\pm 0.42}$ | $15.91_{\pm 0.30}$ | $20.11_{\pm 0.25}$ | $15.40_{\pm 0.85}$ |
| TARGET | $26.53_{\pm 0.23}$ | $3.62_{\pm 0.51}$ | $25.97_{\pm 0.57}$ | $3.08_{\pm 0.27}$ | $25.65_{\pm 1.10}$ | $3.08_{\pm 0.76}$ |
| LANDER | $26.06_{\pm 0.19}$ | $\underline{2.32}_{\pm 0.04}$ | $25.45_{\pm 0.15}$ | $\underline{2.70}_{\pm 0.16}$ | $25.31_{\pm 0.10}$ | $\underline{2.21}_{\pm 0.31}$ |
| FOT | $28.57_{\pm 0.11}$ | $6.31_{\pm 0.11}$ | $28.79_{\pm 0.76}$ | $6.01_{\pm 0.39}$ | $28.33_{\pm 0.56}$ | $4.87_{\pm 0.40}$ |
| FedProTIP (-t) | $\mathbf{28.90}_{\pm 0.39}$ | $7.46_{\pm 0.30}$ | $\mathbf{28.98}_{\pm 1.20}$ | $6.65_{\pm 0.57}$ | $\mathbf{29.32}_{\pm 0.58}$ | $5.70_{\pm 0.51}$ |
| **FedProTIP** | $\underline{28.78}_{\pm 0.08}$ | $\mathbf{1.59}_{\pm 0.20}$ | $\underline{28.06}_{\pm 0.83}$ | $\mathbf{2.10}_{\pm 0.47}$ | $\underline{25.89}_{\pm 0.77}$ | $\mathbf{1.49}_{\pm 0.32}$ |

## D.6 Class-Overlapping Task Variant

We additionally evaluate FedProTIP on a class-overlapping variant of the benchmark, where neighboring tasks share a small number of classes. We consider two settings: (i) one class overlaps across each task

Table 14: Task identity prediction accuracy at each task on 10-split CIFAR100, 6-split DomainNet, and 10-split ImageNet-R at $\alpha = 0.5$.

| Dataset | T1 | T2 | T3 | T4 | T5 | T6 | T7 | T8 | T9 | T10 |
|---|---|---|---|---|---|---|---|---|---|---|
| 10-split CIFAR100 | 1 | 1 | 0.978 | 1 | 1 | 0.989 | 0.997 | 0.989 | 1 | 0.996 |
| 5-split ImageNet-R | 1 | 1 | 0.836 | 0.944 | 0.885 | – | – | – | – | – |
| 10-split ImageNet-R | 1 | 0.895 | 0.940 | 0.912 | 0.878 | 0.906 | 0.943 | 0.904 | 0.896 | 0.874 |
| 20-split ImageNet-R | 1 | 0.9445 | 0.700 | 0.685 | 0.682 | 0.600 | 0.518 | 0.456 | 0.588 | 0.556 |
| (T11-T20) | 0.541 | 0.577 | 0.572 | 0.517 | 0.484 | 0.455 | 0.458 | 0.42 | 0.456 | 0.406 |
| 6-split DomainNet (order 1) | 1 | 1 | 0.673 | 0.88 | 0.8945 | 0.8765 | - | - | - | - |
| 6-split DomainNet (order 2) | 1 | 0.996 | 0.755 | 0.934 | 0.868 | 0.923 | - | - | - | - |

Table 15: Results on a class-overlapping task variant, where neighboring tasks share 1 or 2 classes. We report ACC (%) and FT (%) under oracle task-aware inference, task-agnostic inference with a shared head, and task-agnostic inference with TIP-enabled routing (+TIP) on 10-split CIFAR100 with $\alpha = 0.5$.

| Setting | Task-Aware | | Task-Agnostic | | +TIP | |
|---|---|---|---|---|---|---|
| | ACC | FT | ACC | FT | ACC | FT |
| 1 class overlap | 84.02 | 2.41 | 42.18 | 19.86 | 83.90 | 1.43 |
| 2 class overlap | 82.96 | 3.00 | 28.48 | 9.26 | 76.97 | 5.60 |

boundary (10 classes per task, 11 tasks total) and (ii) two classes overlap (9 classes per task, 14 tasks total) in Table 15.

FedProTIP remains robust under class overlap. With one overlapping class, TIP preserves task-aware performance, achieving 83.90% accuracy and 1.43% forgetting, closely matching the oracle task-aware accuracy (84.02%) while substantially reducing forgetting compared to the shared-head task-agnostic setting. When two classes overlap, task separation becomes more ambiguous, yet TIP still provides strong gains: it improves task-agnostic accuracy from 28.48% to 76.97% while maintaining competitive forgetting (5.60% FT). These results indicate that FedProTIP's subspace-based task identification remains effective even when neighboring tasks are not strictly disjoint.

# E    Experimental Details

## E.1    Datasets

We evaluate our methods and baselines on 3 datasets: CIFAR100, DomainNet, and ImageNet-R. Details on number of classes and dataset division are given in Table 18.

**CIFAR100**    CIFAR100 contains 32×32 sized images from 100 classes, with 600 images per class. In our class-incremental setting, we divide 100 classes into 10 tasks each consisting of 10 classes.

**ImageNet-R**    ImageNet-R (ImageNet-Rendition) (Hendrycks et al., 2021) consists of artistic renditions of 200 object classes from ImageNet, including cartoons, graffiti, and paintings, providing a benchmark for evaluating model's robustness to distribution shifts. In the class-incremental setting, we conduct experiments on ImageNet-R by dividing its 200 classes into 5, 10, and 20 tasks, with each task containing 40, 20, and 10 classes, respectively.

**DomainNet**    DomainNet consists of 224×224 images spanning six visual domains: real, clipart, infograph, painting, quickdraw, and sketch, with each domain treated as a separate task. For training, we sample 10k images per domain, while evaluation uses the full test set. In the main paper, we report results using task ordering 1 (clipart → real → painting → sketch → infograph → quickdraw). For completeness, Table 12 presents results under task ordering 2 (clipart → infograph → painting → quickdraw → real → sketch).

Table 16: Average accuracy (%) across different inference settings. FedProTIP (-t) and FedProTIP correspond to task-agnostic inference without and with task identity prediction, respectively.

| Dataset | Method | Task-Aware | Task-Agnostic |
|---|---|---|---|
| 10-split CIFAR100 | FedAvg | $38.99_{\pm6.52}$ | $15.35_{\pm2.82}$ |
| | GLFC | $75.26_{\pm3.43}$ | $11.86_{\pm2.00}$ |
| | LGA | $85.04_{\pm3.73}$ | $14.35_{\pm1.08}$ |
| | TARGET | $69.81_{\pm1.39}$ | $27.55_{\pm0.89}$ |
| | LANDER | $84.19_{\pm1.94}$ | $37.59_{\pm3.85}$ |
| | FOT | $82.59_{\pm0.61}$ | $41.80_{\pm1.12}$ |
| | **FedProTIP (-t)** | $\mathbf{86.26_{\pm0.27}}$ | $48.41_{\pm0.51}$ |
| | **FedProTIP** | – | $\mathbf{86.00_{\pm0.75}}$ |
| 10-split ImageNet-R | FedAvg | $21.47_{\pm0.11}$ | $8.15_{\pm0.25}$ |
| | GLFC | $30.37_{\pm3.06}$ | $3.18_{\pm1.23}$ |
| | LGA | $41.73_{\pm7.13}$ | $5.76_{\pm1.70}$ |
| | TARGET | $37.64_{\pm0.67}$ | $14.60_{\pm0.66}$ |
| | LANDER | $52.66_{\pm1.29}$ | $23.96_{\pm0.67}$ |
| | FOT | $54.37_{\pm1.35}$ | $26.31_{\pm1.91}$ |
| | **FedProTIP (-t)** | $\mathbf{61.72_{\pm0.04}}$ | $\mathbf{35.64_{\pm0.81}}$ |
| | **FedProTIP** | – | $54.48_{\pm1.87}$ |
| 6-split DomainNet | FedAvg | $10.48_{\pm0.70}$ | $10.72_{\pm0.14}$ |
| | TARGET | $18.11_{\pm0.30}$ | $20.62_{\pm0.18}$ |
| | LANDER | $15.45_{\pm0.58}$ | $21.59_{\pm1.00}$ |
| | FOT | $26.27_{\pm0.43}$ | $24.13_{\pm0.25}$ |
| | **FedProTIP (-t)** | $\mathbf{28.75_{\pm1.45}}$ | $\mathbf{28.85_{\pm1.46}}$ |
| | **FedProTIP** | – | $25.30_{\pm0.30}$ |

Table 17: Metrics (%) of accuracy (↑) and forgetting (↓) computed in the experiments on 10-split CIFAR100. We report the average accuracy and standard deviation over 3 trials, each with different seeds.

| Method | 10-Split CIFAR100 | | | | | |
|---|---|---|---|---|---|---|
| | IID | | $\alpha = 0.5$ | | $\alpha = 0.2$ | |
| | ACC | FT | ACC | FT | ACC | FT |
| FedAvg | $18.92_{\pm2.45}$ | $63.20_{\pm1.36}$ | $15.35_{\pm2.82}$ | $62.90_{\pm0.79}$ | $15.76_{\pm7.28}$ | $52.80_{\pm4.36}$ |
| GLFC | $14.07_{\pm1.10}$ | $69.17_{\pm0.31}$ | $11.86_{\pm2.00}$ | $68.20_{\pm2.36}$ | $10.33_{\pm1.97}$ | $63.98_{\pm1.96}$ |
| LGA | $14.93_{\pm1.09}$ | $72.06_{\pm1.44}$ | $14.35_{\pm1.07}$ | $71.09_{\pm2.56}$ | $11.67_{\pm0.65}$ | $65.82_{\pm1.20}$ |
| TARGET | $29.56_{\pm0.75}$ | $42.73_{\pm4.95}$ | $27.37_{\pm1.00}$ | $37.60_{\pm5.30}$ | $23.05_{\pm2.56}$ | $34.63_{\pm2.74}$ |
| LANDER | $39.09_{\pm1.99}$ | $\underline{9.27}_{\pm1.11}$ | $37.59_{\pm3.85}$ | $\underline{10.21}_{\pm1.59}$ | $23.56_{\pm5.61}$ | $\underline{13.28}_{\pm4.17}$ |
| FOT | $46.86_{\pm2.67}$ | $21.11_{\pm0.87}$ | $41.80_{\pm1.12}$ | $20.86_{\pm1.12}$ | $34.65_{\pm1.39}$ | $18.09_{\pm0.72}$ |
| FedProTIP (-t) | $\underline{52.30}_{\pm1.81}$ | $15.66_{\pm0.77}$ | $\underline{48.41}_{\pm0.51}$ | $15.59_{\pm0.80}$ | $\underline{42.19}_{\pm0.97}$ | $14.91_{\pm1.11}$ |
| **FedProTIP** | $\mathbf{87.94_{\pm0.79}}$ | $\mathbf{0.34_{\pm0.59}}$ | $\mathbf{86.00_{\pm0.75}}$ | $\mathbf{0.83_{\pm0.47}}$ | $\mathbf{81.94_{\pm1.02}}$ | $\mathbf{1.35_{\pm0.47}}$ |

## E.2 Model Architecture

We use a ResNet-18 (He et al., 2016) pre-trained on ImageNet-1K as the backbone network for all datasets in the main paper. After learning the first task, we freeze the first two residual blocks of ResNet and only update the remaining parts of the model. At the end of each task, the parameters of the last fully connected layer are extended by adding neurons as classes are incremented. In addition, while learning new tasks we freeze the parameters of the last fully connected layer corresponding to previously learned tasks.

## E.3 Training Details

In all experiments, we use the SGD optimizer with a learning rate of 0.01 and a weight decay of $5 \times 10^{-4}$ for all baselines. Unless otherwise stated, the batch size is set to 64. For training, the local epoch is fixed at 5 and the number of global rounds per task is 50 for CIFAR100 and ImageNet-R, and 20 for DomainNet. To

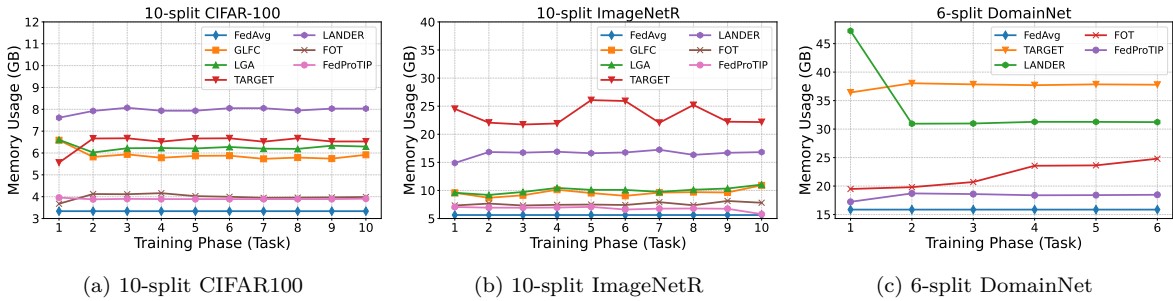

Figure 7: GPU memory usage (GB) on a single NVIDIA H200 GPU. We report the maximum GPU memory allocated at each training phase.

Table 18: Dataset details used in experiments.

| Dataset | # Classes | # Tasks | # Train | # Test |
|---------|-----------|---------|---------|--------|
| CIFAR100 | 100 | 10 | 50,000 | 10,000 |
| DomainNet | 345 (per domain) | 6 | 60,000 | 20,674 |
| ImageNet-R | 200 | 5/10/20 | 67,080 | 19,464 |

maintain consistent number of selected clients across different experiments, we apply a client fraction 1.0 at each round for 5 clients, and 0.5 and 0.25 for 10 and 20 clients, respectively. We set the threshold $\epsilon_l = 0.7$ for all datasets. Additional ablation study on the threshold value is provided in Appendix D.4. We describe training details for each baseline in the following.

**GLFC**   GLFC (Dong et al., 2022) employs exemplar replay by storing a subset of raw samples for each task. For CIFAR100, following the original paper we set the memory size to 2000; to satisfy memory constraints, for DomainNet and ImageNet-R the memory size is limited to 1000. GLFC incorporates sample reconstruction optimization to select the best old model on a proxy server, where the selected model is used in the next task via distillation. For this optimization we use the L-BFGS optimizer with a learning rate of 0.5 for CIFAR100 and DomainNet, and 0.1 for ImageNet-R.

**LGA**   LGA (Dong et al., 2023) extends GLFC by relying on a gradient encoding model to reconstruct perturbed images from the gradients received on a proxy server. Additionally, it introduces self-supervised prototype augmentation to enhance selection of the best old model from the reconstructed perturbed prototype images. In our experiments, we use LeNet as the gradient encoding model for all datasets, and the SGD optimizer to generate perturbed images. We retain the same experimental settings as implemented by GLFC if the two approaches share the same configurations.

**TARGET**   TARGET (Zhang et al., 2023) leverages the previously trained global model to distill knowledge from past tasks into the current model while also training a generator to produce synthetic data that captures global information from previous tasks. In our implementation, we use 8k synthetic samples with a batch size of 256 for CIFAR100, following the original paper's hyperparameters for generator training rounds, distillation schedules, and learning rates. For DomainNet, we generate 12,800 synthetic samples in batches of 64, with 200 rounds of data generation and 100 generator iterations per round. For ImageNet-R, we use 12,800 synthetic samples with a batch size of 64, and set the data generation process to 40 rounds with 40 generator iterations per round to fit GPU memory constraints.

**LANDER**   LANDER (Tran et al., 2024) utilizes label text embeddings (LTE) generated by pretrained language models as anchor points, constraining feature embeddings of the training data around the corresponding class LTEs. Additionally, these anchors guide the generator optimization, ensuring that the global model embeddings of synthetic samples remain close to LTEs, thereby generating more meaningful samples. We follow the same experimental settings and use the provided LTEs for LANDER on CIFAR100 as suggested in the original paper. For other datasets, since the official implementation does not include LTE generation,

we construct the LTE pool using a pretrained CLIP model (Radford et al., 2021). We adopt the same prompt template, "A photo of a class", where `class` denotes the label of each class. For DomainNet and ImageNet-R, we match the number of synthetic samples and data generation procedure used in TARGET, while keeping all other configurations consistent with CIFAR100.

**FOT**   FOT (Bakman et al., 2024) adapts GPM to the FCL setting, with key differences from FedProTIP occurring at the end of each task: (i) A client transmits its input representation multiplied by a standard normal vector with a predefined sampling dimension; (ii) the randomized input representations are averaged and the core bases of the gradient subspace are extracted from these aggregated representations; and (iii) the global model parameters are updated via orthogonal projection using these bases on the server side. In our implementation, for each dataset we set the sampling dimension of the standard normal vector to five times the feature size. As for the threshold required to obtain bases from the aggregated features, we use the starting value of 0.87 with an increment of 0.01 for each new task for CIFAR100, while for DomainNet and ImageNet-R we use threshold 0.9 with an increment of 0.01.

### E.4   Data Heterogeneity

To assess the impact of data heterogeneity on FCL systems, we partition dataset across clients based on the heterogeneity level controlled by the Dirichlet distribution. For an IID split, we randomly shuffle the dataset indices and divide them into equal-sized subsets, ensuring each client receives a uniform share of the dataset, independent of class labels. This ensures balanced data distribution across the clients. For a non-IID split, we control heterogeneity using the Dirichlet distribution parameterized by $\alpha$. Specifically, for each class, we sample a probability vector from $Dir(\alpha)$ to determine the proportion of data assigned to each client. We prevent empty assignments, guaranteeing that each client holds at least one sample from every class present in its assigned task. Smaller values of $\alpha$ lead to a more skewed distribution, creating more severe class imbalance across clients.

## F   Discussions

### F.1   Related Works on FCL

Federated continual learning (FCL) tackles the challenge of continuously learning from decentralized data while maintaining knowledge across tasks. An early approach to FCL, FedWeIT (Yoon et al., 2021), decomposes parameters into task-generic and task-specific ones, focusing on a task-incremental setting where the task ID is known during inference. More recently, TagFed (Wang et al., 2024a) introduces a model extraction-based approach that mitigates forgetting by maintaining task-specific sub-networks with parameter masks, selectively updating recurring tasks while employing group-wise knowledge aggregation to cluster clients based on feature-based distillation at the server. pFedDIL (Li et al., 2025) propose a personalized federated domain-incremental learning method that estimates task correlations using an auxiliary classifier to determine whether to reuse a previous model or train a new one, with final predictions obtained through a weighted ensemble of personalized models.

In the realm of replay-based methods, CFeD (Ma et al., 2022) employs knowledge distillation enabled by a surrogate dataset made available to clients as well as the server. GLFC (Dong et al., 2022; 2023) addresses catastrophic forgetting by leveraging class-aware gradient compensation and class-semantic relation distillation, while relying on the memory of old examples. The follow-up studies (Liu et al., 2023; Dai et al., 2024; Li et al., 2024c;a) reduce the size of the replay cache but remain reliant upon old samples.

To address the reliance on real data, generative model-based FCL methods have been proposed. FedCIL (Qi et al., 2023) employs a GAN with an auxiliary classifier to enable generative replay, preventing forgetting and aggregating global knowledge across clients. TARGET (Zhang et al., 2023) and MFCL (Babakniya et al., 2024) introduce data-free knowledge distillation that enables the use of synthetic examples to transfer knowledge from an old global model to client models. LANDER (Tran et al., 2024) builds on this by incorporating label text embeddings from pretrained language models as anchors, generating more meaningful samples and further improving the ability to mitigate forgetting. AF-FCL (Wuerkaixi et al., 2024) leverages

feature generative replay with a normalizing flow (NF) model to estimate the probability density of generated features, enabling deliberate forgetting of biased features caused by data heterogeneity.

While many prior works have reported results in the class-incremental learning (CIL) setting, where task IDs are unknown during inference, they still exhibit substantial performance degradation compared to the relatively easier task-incremental learning (TIL) scenario. In contrast, FedProTIP enables accurate task-ID prediction for each test sample, thereby achieving near-TIL performance even under the more challenging CIL setting.

**Parameter-efficient federated fine-tuning.** A related but distinct line of work studies parameter-efficient adaptation of pretrained models, most commonly through low-rank adapters. In continual learning, methods such as O-LoRA (Wang et al., 2023) and InfLoRA (Liang & Li, 2024) design task-specific LoRA update subspaces to reduce cross-task interference. More directly related to our setting are recent LoRA-based approaches for federated continual learning, such as PILoRA (Guo et al., 2024), which learns incremental LoRA modules for federated class-incremental learning, and LoRM (Salami et al., 2025), which studies closed-form merging of parameter-efficient modules for federated continual learning. Beyond continual learning, federated PEFT methods such as FLoRA (Wang et al., 2024b) and FlexLoRA (Bai et al., 2024) focus on stable LoRA aggregation under heterogeneous client resources or adapter ranks, while recent methods including FedSVD (Lee et al., 2025) and FedRot-LoRA (Zhang et al., 2026) exploit low-rank geometry through adaptive SVD-based reparameterization or rotational alignment before aggregation. These PEFT-based approaches are complementary to FedProTIP rather than direct substitutes: they are especially attractive when the backbone is very large and mostly frozen, whereas FedProTIP targets settings where the full model is updated and the per-layer projection overhead remains manageable.

### F.2 Privacy Considerations in FedProTIP

**FOT's privacy mechanism.** FOT does not transmit raw activations. Each client $k$ computes a randomized sketch $A_k = \sum_j x_{k,j}^* (g_j^\ell)^\top$, where $g_j^\ell \sim \mathcal{N}(0, I_{s_\ell})$, and the sketches are summed via secure aggregation (SecAgg) (Bonawitz et al., 2017). The server observes only $A = \sum_k A_k$ and extracts the global subspace by SVD. FOT's privacy argument relies on SecAgg hiding individual $A_k$ and on the randomized aggregate sketch used for distributed subspace estimation.

**Privacy limitation of per-client basis transmission.** In the default FedProTIP protocol, each client transmits bases $U_k^{(t)} \in \mathbb{R}^{d_l \times r_k}$ to the server. These bases reveal client-specific principal directions of the local feature space and may expose information about the client's task distribution. Since bases are sent per client, the server can attribute specific subspace directions to individual clients. To mitigate this limitation, we introduce a secure-aggregation-compatible extension of FedProTIP below.

**Weighted Gaussian sketch.** We introduce a client-side sketch that makes FedProTIP compatible with SecAgg while preserving aggregate subspace recovery. Each client performs SVD locally, then computes

$$B_k = U_k \operatorname{diag}(\sigma_k) G_k \in \mathbb{R}^{d_l \times s_l}, \tag{41}$$

where $G_k \sim \mathcal{N}(0, I)$ is an independent per-client random matrix and $\sigma_k$ denotes the retained singular values. The server receives only the secure aggregate $B = \sum_k B_k$ via SecAgg. This construction has three useful properties:

- Individual client bases are not exposed to the server.

- The per-client $G_k$ randomizes directions within each sketch, obscuring the original basis coordinates.

- The aggregate can be written as $B = M\tilde{R}$, where $M = [U_1 \Sigma_1, \ldots, U_K \Sigma_K]$ and $\tilde{R}$ is a block-diagonal Gaussian matrix, aligning the construction with the randomized-sketching principle used by FOT.

Table 20 summarizes the communication payload, SecAgg compatibility, and server-side observability of each method.

**Empirical validation.** Table 19 verifies that the sketch preserves accuracy on 10-split CIFAR100. The sketch dimension $s_l$ controls the trade-off between performance and communication cost. With $s_l = 128$, FedProTIP achieves near task-aware accuracy with TIP (85.60% vs. 85.76%) while reducing per-client communication from 366.9 MB/task for the full-dimensional sketch ($s_l = d_l$) to 16.1 MB/task, with compatibility with SecAgg.

Table 19: Effect of weighted Gaussian sketch on FedProTIP. 10-split CIFAR100, $\alpha = 0.5$, $\epsilon_l = 0.775$, 5 clients. Per-client communication is reported per task.

| Sketch $s_l$ | Task-Aware | | +TIP | | Per-client |
| | ACC | FT | ACC | FT | MB/task |
| --- | --- | --- | --- | --- | --- |
| $d_l$ | 85.76 | 1.31 | 85.17 | 0.21 | 366.9 |
| 128 | 86.25 | 2.79 | 85.60 | −1.47 | 16.1 |
| 64 | 85.42 | 3.24 | 75.36 | 11.55 | 8.1 |

Table 20: Privacy comparison of subspace communication methods. $d_l$: layer width, $r_k$: retained rank, $s_l$: sketch dimension.

| Method | Client sends | SecAgg | Server observes |
| --- | --- | --- | --- |
| FOT | $X_k^* G$ $(d_l \times 5d_l)$ | ✓ | $\sum_k A_k$ |
| FedProTIP | $U_k$ $(d_l \times r_k)$ | ✗ | Each $U_k$ |
| FedProTIP (sketch) | $U_k \Sigma_k G_k$ $(d_l \times s_l)$ | ✓ | $\sum_k B_k$ |

