# OpenReview forum: "FedProTIP: Task-Agnostic Federated Continual Learning via Replay-Free Gradient Projection"
_TMLR — Accepted by TMLR_

### Review · Reviewer_P8ee · 2026-03-26

**Summary Of Contributions:**

## Summary
This paper introduces FedProTIP, or Federated Projection-based Continual Learning with Task Identity Prediction. This framework for federated continual learning reduces catastrophic forgetting without needing replay buffers or generative models. Each client extracts low-rank core bases from intermediate activations via randomized SVD, transmits only these compact bases to the server, which aggregates them into a global orthonormal subspace. During training, clients apply projected gradient descent (PGD) locally, constraining updates to the orthogonal complement of previously learned subspaces. At inference, a task identity prediction (TIP) mechanism uses subspace relevance alignment to estimate task identity without access to ground-truth task labels, enabling task-agnostic operation. Experiments on CIFAR-100, ImageNet-R, and DomainNet show consistent gains over six baselines in both accuracy and forgetting metrics, with lower training time, memory, and communication overhead.



## Key Strengths
- The proposed framework integrates gradient projection directly into local client optimization rather than only at the server post-aggregation, which is a meaningful design choice that is both theoretically analyzed and empirically validated.
- The TIP mechanism is a practical and efficient solution to the task-agnostic inference problem that reuses previously computed subspace information at negligible cost.
- Comprehensive empirical evaluation covering multiple datasets, heterogeneity levels, backbone architectures, and ablation conditions.



## Key Weaknesses
- The TIP mechanism relies on cosine similarity between a test relevance vector and stored reference vectors. The paper does not provide a theoretical characterization of when TIP is expected to succeed or fail, nor a bound on task identification error rate.
- The theoretical bounds in Theorems 1 and 2 depend on $\rho_t$ and $\beta$, which are defined in gradient space and are not directly controllable by the practitioner. The paper acknowledges this gap but provides no guidance on how these quantities behave in practice as a function of the SVD threshold $\epsilon_l$ or dataset properties.
- The global subspace aggregation (Eq. 8) depends on the order. The server adds client bases one at a time. The final orthonormal basis can vary based on the order in which the clients are processed. This possible inconsistency is not discussed.

**Audience:**

Yes

**Audience Explanation:**

FedProTIP tackles a real limitation of earlier methods, which only projected after aggregation. It provides an interesting and meaningful solution. The mix of local projection, efficient subspace communication, and task-agnostic inference creates a clear and complete contribution. The result that near task-aware accuracy can be recovered purely from subspace geometry (e.g., 86.00% task-agnostic vs. 86.26% task-aware on CIFAR-100) is an interesting finding that the community will find informative. The paper is also of interest to researchers working on privacy-preserving ML, as the architecture avoids transmitting raw data or activations.

**Claims And Evidence:**

No

**Claims Explanation:**

- The theoretical claims are sound but incomplete. The bounds are stated in terms of unobservable quantities ($\rho_t$, $\beta$), and no empirical estimation or proxy for these quantities is provided. The forgetting bound in Eq. 14 scales as $\mathcal{O}\left(T \bar{\beta}_{T} \sqrt{E}\right)$, which suggests forgetting grows with $T$ unless $\bar{\beta}_T$ shrinks proportionally, but there is no discussion of whether this regime is achieved in practice.
- The privacy discussion in Appendix F.2 is largely qualitative. The claim that transmitting low-rank bases reduces gradient inversion risk compared to raw activations is plausible but not formally demonstrated. A reference to the cited work (Zhang et al., 2025) is given, but no quantitative privacy analysis is conducted for FedProTIP itself.

**Requested Changes:**

1- The paper should include a direct analysis or empirical measurement of TIP accuracy (i.e., the fraction of test samples correctly assigned to the right task head), separated from end-to-end classification accuracy. This would make it possible to diagnose whether performance degradation in harder settings (e.g., 20-split ImageNet-R) stems from worse task routing or from the projection mechanism itself.

2- The order-dependence of the global subspace aggregation (Eq. 8) should be explicitly discussed. Is the span of $\Phi^{(t)}$ invariant to client ordering? If so, Proposition 1 in Appendix C should be stated earlier and more prominently. If not, the sensitivity to client ordering should be assessed empirically.

3- Theorems 1 and 2 should be accompanied by at least an empirical estimation of $\rho_t$ and $\beta$ over the course of training on one of the benchmark datasets. Even a single illustrative plot would substantially strengthen the connection between theory and practice and help readers understand whether the bounds are tight or loose.

4- The paper should clarify whether the incremental threshold schedule ($\epsilon$ increasing by $0.001$ per task) is theoretically motivated or purely heuristic. If heuristic, the sensitivity to this schedule should be evaluated alongside the static threshold ablation in Figure 4 and Table 10.

---

> ### Author Response · Authors · 2026-04-26
> **Response to Reviewer P8ee [1]**
>
> ## Summary of Changes
>
> - **TIP accuracy (P.1):** We now report standalone TIP routing accuracy in Figure 4(a) and Table 14, and clarify in Section 6.1 how routing quality contributes to the gap between FedProTIP and FedProTIP(-t).
>
> - **Order-invariance (P.2):** We direct the reviewer to Proposition 1 (Appendix C), showing that
> $\mathrm{span}(\Phi_l^{(t)}) = \mathrm{span}(\bigcup_k U_{l,k}^{(t)})$
> regardless of client processing order, and reference this result after Eq. (9).
>
> - **Empirical $\hat{\rho}_t$ and $\hat{\beta}$ (P.3):** We added Figure 2, which reports both quantities across tasks and thresholds on 10-split CIFAR100, and revised Section 5 to present the $\epsilon_l$ connection as an empirical diagnostic.
>
> - **Threshold schedule (P.4):** We now explicitly describe the incremental $\epsilon_l$ schedule as a heuristic and provide guidance for longer task sequences.
>
> - **Privacy (P.5):** Appendix F.2 has been replaced with a secure aggregation-compatible weighted Gaussian sketch, supported by a privacy comparison table (Table 20) and an empirical ablation (Table 19). Please see R.1 for details.
>
> ---
>
> ### P.1 Separate TIP accuracy metric.
>
> **Response:** Task identity prediction accuracy is now reported as a standalone metric in Figure 4(a) and Table 14 (Appendix D). On 10-split CIFAR100, TIP achieves near-perfect routing accuracy ($\geq 0.978$) across all tasks. On 10-split ImageNet-R, TIP accuracy remains above $0.87$, while on 6-split DomainNet it remains above $0.67$ despite the shared label space. On 20-split ImageNet-R, routing accuracy decreases in later tasks (T11–T20) as the number of classes and examples per task decreases. FedProTIP nevertheless remains above all baselines in Table 2, indicating that local projection contributes substantial gains even when task routing becomes less reliable. Section 6.1 now references these results directly.
>
> ---
>
> ### P.2 Order-dependence of global subspace aggregation (Eq. 9).
>
> **Response:** The actual basis can depend on processing order, but the resulting span is invariant. Proposition 1 (Appendix C) proves that, after orthogonalization,
> $\mathrm{span}(\Phi_l^{(t)}) = \mathrm{span}(\bigcup_{k=1}^K U_{l,k}^{(t)})$
> regardless of client order. Since the projection operator
> $P^{(t)} = I - \Phi^{(1:t)}(\Phi^{(1:t)})^\top$
> depends only on the column space, the projected updates are order-invariant up to numerical effects.
>
> ---
>
> ### P.3 Empirical estimation of $\rho_t$ and $\beta_\tau^{(t-1)}$.
>
> **Response:** We now provide empirical measurements on 10-split CIFAR100 ($\alpha = 0.5$). Motivated by Assumptions 4 and 5, we estimate these quantities at synchronization iterates,
>
> $$
> \hat\rho_t = \frac{\\|P^{(t-1)}\nabla{L}^{(t)}(W)\\|^2}{\\|\nabla{L}^{(t)}(W)\\|^2}, \qquad
> \hat\beta_{\tau \to t} =
> \frac{\\|P^{(t-1)}\nabla{L}^{(\tau)}(W)\\|}{\\|\nabla{L}^{(\tau)}(W)\\|},
> $$
>
> averaged across communication rounds. Results are reported in Figure 2.
>
> **(a) $\hat{\rho}_t$ across the task sequence.**
> $\hat{\rho}_t$ decreases with $\epsilon_l$, from $\approx 0.47$ at $\epsilon_l = 0.1$ to $\approx 0.05$ at $\epsilon_l = 0.9$. This is consistent with the role of $\rho_t$ in Theorem 1: a higher threshold enlarges the protected subspace, reducing the gradient energy available for current-task descent and increasing the $1/\rho_t$ prefactor. Within each threshold, $\hat{\rho}_t$ remains approximately flat across tasks, suggesting that $\Phi^{(1:t-1)}$ remains low-rank relative to $d_l$ over the measured sequence.
>
> **(b) $\hat{\beta}_{\tau \to t}$ heatmap.**
> At the default threshold $\epsilon_l = 0.7$, values range from $0.18$ to $0.32$ and are approximately uniform across task pairs, suggesting no disproportionate interference between any particular pair. This moderate leakage is consistent with the low empirical forgetting (Table 1).
>
> **(c) $\hat{\rho}_t$ and $\hat{\beta}$ vs. $\epsilon_l$.**
> Both quantities decrease monotonically with $\epsilon_l$. Although $\rho_t$ and $\beta$ are defined in gradient space, this trend shows that $\epsilon_l$ provides an empirical handle on the trade-off between stability and plasticity. The same observation helps explain the relative insensitivity of accuracy to $\epsilon_l$ (Table 11): lower $\hat{\rho}_t$ suggests slower current-task convergence, while lower $\hat{\beta}$ suggests reduced forgetting, so the two effects can partially offset each other. At $\epsilon_l = 0.7$ with $T = 10$ and $E = 50$, we observe $\hat{\rho}_t \approx 0.13$ and $T\bar{\beta}_T \approx 2.4$, which is consistent with the strong accuracy and low forgetting observed in Table 1. For longer task sequences, the measured quantities could be used to guide a larger or adaptive $\epsilon_l$ schedule, trading slower per-task convergence for reduced interference.
>
> ---

---

> > ### Author Response · Authors · 2026-04-26
> > **Response to Reviewer P8ee [2]**
> >
> > ### P.4 Is the threshold schedule theoretically motivated or heuristic?
> >
> > **Response:** The incremental schedule ($\epsilon_l$ increasing by $0.001$ per task) is **heuristic**, following FOT's convention. The underlying intuition is a scaling concern: as more tasks accumulate, there are more past representations that each new update could interfere with. Increasing $\epsilon_l$ is intended to counter this by retaining more activation directions in the protected subspace, forcing each new task's update into a more constrained region. Figure 2(c) quantifies the effect: higher $\epsilon_l$ reduces $\hat{\beta}$ (less interference) at the cost of lower $\hat{\rho}_t$ (slower learning). The schedule thus trades plasticity for reduced interference across a growing set of past tasks.
> >
> > The increment of $0.001$ per task is conservative ($\Delta\epsilon = 0.01$ over 10 tasks), consistent with the insensitivity observed in Table 11. For longer task sequences, a larger increment or an adaptive schedule calibrated to the measured $\hat{\rho}_t$ and $\hat{\beta}$ may be preferable; we leave this to future work.
> >
> > ---
> >
> > ### P.5 Privacy discussion.
> >
> > **Response:** We agree that the original Appendix F.2 was too qualitative. As detailed in R.1, we replaced it with a concrete secure-aggregation-compatible mechanism, i.e., the weighted Gaussian sketch ($B_k = U_k\Sigma_kG_k$). The revised Appendix F.2 includes (i) a formal description of the sketch construction and its subspace recovery property, (ii) a privacy-relevant communication comparison table (Table 20), and (iii) an empirical ablation (Table 19) showing that $s_\ell=128$ preserves accuracy while requiring only 16.1 MB/task per client. We present the sketching mechanism as a practical way to reduce the privacy risk of sending per-client bases, rather than as a complete formal privacy guarantee.

---

### Review · Reviewer_Ri81 · 2026-03-30

**Summary Of Contributions:**

The submission proposes a new method at the intersection of continual and federated learning: edge/client devices train on their own private data across changing data distributions, sharing a low rank task basis with a centralized server that is used to share learning across clients without sharing data. Furthermore, the submission proposes a new way to identify the task being solved without access to task labels. The submission characterizes formally the stability-plasticity tradeoff of the method, and empirically validates it on a variety of standard benchmarks against a set of prior methods. Furthermore, ablations are provided demonstrating how the method interacts with client heterogeneity, task labeling, and locality of gradient projections.

**Additional Comments:**

One puzzle I noticed is that on DomainNet -t wins on ACC but +t wins on FT, i.e. the task ID helps the model forget less but makes accuracy worse across all heterogeneity levels. Perhaps it is just noise, but I wonder if that interacts in some interesting way with TIP's dependence on input distribution changes for task identification.

**Audience:**

Yes

**Audience Explanation:**

Continual learning and federated learning are both core ML areas and the contribution fits solidly in TMLR's wheelhouse -- no concerns there. The extension combining GPM and FL by sharing per-client principal subspaces only is straightforward but cleanly done. I think if some of the issues above are addressed this would be a useful addition to the literature.

**Broader Impact Concerns:**

No significant concerns, but I do think the paper should engage with the security aspects of FL more strongly.

**Claims And Evidence:**

No

**Claims Explanation:**

I think on the whole I find the submission generally well-written, the motivation of the method clear and easy to understand (local continual learning constraints, less communication overhead, etc), and the performance good as well. I am less sure that I am fully convinced by the impact of the theorems: while bounds are provided I'm not entirely clear if they are strong in some abstract sense or concretely better than prior work.

Furthermore, I have concerns about both informal benefit and empirical performance claims relative to the nearest competitor method (FOT), as well as the generality of the TIP extension. I believe these can be addressed straightforwardly in revision but as the paper stands I think it falls short of the TMLR evidence rubric.

Specifically:
1. The paper claims that FOT shares layer wise intermediate activations with the server, which are used to extract core bases. From a brief review of FOT, that method uses secure aggregation and a randomized global PCA that has formal differential privacy guarantees (i.e. clients share not the raw activations but layer inputs multiplied by a random matrix, which the server uses to reconstruct the principal subspace in a distributed variant of the Johnson-Lindenstrauss transform). To the extent that the current contribution takes issue with those guarantees, it should take care to be precise and correct about what the prior method does, as well as what its issue is with the privacy risks. Furthermore, the submission should be clearer on where it mitigates such attacks and provides additional privacy. Even if FOT provided raw activations to the server: supposing we go from the server accessing raw activations to accessing the most informative directions, doesn't this reduce the search space for model inversion attacks? It might make the attack less accurate (because signal dimensions are dropped) or more accurate (because noise dimensions are dropped and dimensionality is reduced). My prior is on the latter and I don't think the paper defends the former. If instead we compare randomized client activations sent by secure aggregation (what FOT actually seems to do) to non-randomized principal directions sent in the clear (the current contribution), the picture becomes even less favorable. There's also a separate question of what a client can learn about other clients from the shared global feature subspace, a risk that exists for FedProTIP but not methods which do not send anything but global weights to the client. For example, can a client track changes in the global feature subspace to infer the task distribution of the other clients, and is this a potential concern?
2. I'm not sure what to make of the theoretical analysis. Bounds are given but are hard to understand without significant context given in the appendix, and even with that context are not contextualized. Do we care about these bounds? Are there tighter than prior methods or in some other way meaningful?
3. The current paper claims FOT performance on 10-split CIFAR100 at 46.86% and forgetting at 21.11%. The FOT paper's table 1 reports 10-split CIFAR100 numbers of 71.9%% accuracy and 0.87% forgetting. As far as I can tell, the definition of the forgetting metric is different between the two papers (and the present contribution's metric correctly matches what is in Chaudhry et al. 2018), so that explains the forgetting difference. But I could not determine what would explain the accuracy difference -- could the authors address this?
4. I am confused about how task identification with access to inputs only (no labels) could work in the general case. With a domain-incremental setup, the input distribution changes, so tasks can be identified from inputs alone. With a class-incremental setup of the split-CIFAR or split-ImageNet type, the input distribution still changes alongside the added classes. Suppose instead I have class-incremental setup with image data and one task is animacy (animate / inanimate) and the other size (large / small object). By construction, nothing will change about the input distribution (I play back the same data epoch twice). How would TIP distinguish between the two tasks in this case?

**Requested Changes:**

Of the changes suggested above, I think the following are critical:
1. Addressing privacy claims relative to prior work, and/or providing more precise claims of exactly what attacks the current contribution mitigates.
2. Clarifying the performance differences for FOT reported here vs in the original paper.

Some remaining changes are strongly recommended:
3. Better contextualizing the formal bounds, and / or discussing what their role is in either suggesting practical decisions and consequences, or motivating future directions or research. Basically, what's the point of these bounds? I think pulling a little more of the informal intuition behind the theorems from the appendix would also help here.
4. Moderating the TIP claims if indeed TIP requires that input distributions change in the class-incremental setting.

I also have some minor suggestions that might help with clarity:
5. I appreciate the discussion of complexity in section 6.5. But I think it mixes two things: wall times and memory usage (which depend on implementation details of the underlying software), and complexity (which does not). It might help to separate the two, and / or add a table comparing memory, computation, and communication cost by method in an apples-to-apples way. Notably, I think "per layer" or "per step" big-O may hide some significant costs that prevent this style of method from scaling to larger networks (where the number of layers and / or training steps might be substantially larger). Also on the front of scalability / complexity, if the paper takes seriously the claim of improved complexity (memory / time / space, etc), it starts engaging with parameter-efficient FCL methods (FLoRA, FLoRA-NA, FlexLoRA, and others). These aren't discussed and engaged with. I don't think it's critical given the paper's focus outside the PEFT side of the FCL space, but if reduced communication cost and memory usage is a selling point, there are significant competitors in that literature.

---

> ### Author Response · Authors · 2026-04-26
> **Response to Reviewer Ri81 [1]**
>
> ## Summary of Changes
>
> - **Privacy and FOT characterization (R.1):** We corrected the description of FOT and removed statements implying that FOT transmits raw activations. We also added a secure-aggregation-compatible variant of FedProTIP based on weighted Gaussian sketches, $B_k = U_k\Sigma_kG_k$, with empirical validation and a privacy comparison (Appendix F.2).
>
> - **Theoretical claims (R.2):** We rephrased the claims and added empirical estimates of $\hat\rho_t$ and $\hat\beta_{\tau\to t}$ across five thresholds (Figure 2). The revised Section 5 now presents the connection to $\epsilon_l$ as an empirical diagnostic rather than a rigorous consequence of the bounds.
>
> - **Evaluation clarity and TIP limitations (R.3, R.4, R.6):** We clarified that Table 1 uses task-agnostic inference, added standalone TIP accuracy in Figure 4(a) and Table 14, discussed TIP's assumptions and limitations in Section 4.4, and explained the DomainNet trade-off in Section 6.2.
>
> - **Complexity and PEFT context (R.5):** We separated asymptotic complexity from empirical wall-time and memory measurements in Section 6.5, and added discussion of parameter-efficient FCL methods in Appendix F.1.

---

> > ### Author Response · Authors · 2026-04-26
> > **Response to Reviewer Ri81 [2]**
> >
> > ### R.1 Privacy claims relative to FOT.
> >
> > **Response:** We thank the reviewer for bringing up this point. We agree that the original manuscript overstated the privacy contrast with FOT and incorrectly described aspects of its communication protocol. We have corrected the manuscript accordingly and revised our claims to distinguish the following points: (i) what FOT actually transmits, (ii) what privacy limitations remain in the original FedProTIP basis-sharing design, and (iii) how the new sketching variant makes FedProTIP compatible with secure aggregation.
> >
> > **Correction on FOT.**
> > FOT does not transmit raw activations. Each client computes a randomized sketch $A_k = \sum_j x_{k,j}^* (g_j^\ell)^\top$ with $g_j^\ell \sim N(0, I_{s_\ell})$, and the sketches are summed via secure aggregation (SecAgg). The server observes only the aggregate $A = \sum_k A_k$ and extracts the global subspace by SVD. Thus, FOT's privacy argument relies on SecAgg hiding individual client sketches and on the randomized aggregate sketch used for distributed subspace estimation. We have corrected all manuscript statements that previously described FOT as transmitting raw activations.
> >
> > **Limitation of transmitting per-client bases.**
> > We agree that transmitting non-randomized per-client bases $U_k$, as in the original FedProTIP design, reveals client-specific principal directions and is therefore less privacy-preserving than FOT's secure-aggregation protocol. In particular, because the bases are sent separately by each client, the server may attribute subspace directions to individual clients. We now explicitly identify this as a limitation of the original design and address it with the sketching variant described below.
> >
> > **Secure aggregation compatible weighted Gaussian sketch.**
> > We introduce a client-side sketch that makes FedProTIP compatible with SecAgg while preserving the ability to recover aggregate subspace. Each client performs SVD locally, then computes
> > $B_k = U_k \operatorname{diag}(\sigma_k) G_k \in \mathbb{R}^{d_\ell \times s_\ell}$,
> > where $G_k \sim N(0,I)$ is an independent per-client random matrix. The server receives only the secure aggregate $B=\sum_k B_k$. This modification has three effects: (i) individual client bases are no longer exposed to the server, (ii) the random projection obscures the basis coordinates within each client sketch, and (iii) the aggregate can be written as $B=M\widetilde{R}$, where $M=[U_1\Sigma_1,\ldots,U_K\Sigma_K]$ and $\widetilde{R}$ is a stacked Gaussian matrix, aligning the construction with the randomized sketching used by FOT. Table 19 verifies that this sketch preserves accuracy on 10-split CIFAR100. At $s_\ell=128$, task-aware accuracy (86.25%) and +TIP accuracy (85.60%) closely match the full-sketch baseline ($s_\ell=d_\ell$), with per-client communication of 16.1 MB/task, substantially lower than FOT's $\sim$367 MB/task at $s_\ell=5d_\ell$. At $s_\ell=64$, subspace recovery degrades noticeably (+TIP ACC drops to 75.36%), indicating that the sketch dimension must be sufficiently large relative to the retained ranks for faithful recovery.
> >
> > **Residual leakage from the global subspace.**
> > A client receiving $\Phi^{(t)}$ can observe which directions were added relative to $\Phi^{(t-1)}$, potentially revealing information about the aggregate task distribution. We acknowledge this residual leakage. However, the exposed object is an orthonormal basis — a set of directions, transmitted without the accompanying singular values or per-sample coordinates. Reconstructing an activation $a$ from such a basis $U$ requires the projection coefficients $U^\top a$, which are never transmitted; the bases by themselves therefore cannot be inverted to recover individual samples or training data. While this does not constitute a formal privacy guarantee, since the directions themselves can reveal coarse structural properties of the underlying activation distribution, it limits per-sample reconstruction even in the original basis-sharing design. Under suggested SecAgg-compatible sketch, the added directions further reflect an aggregate over participating clients rather than identifiable per-client bases. The influence of any single client is reduced as the number of participating clients grows, and the energy threshold retains only directions with sufficient aggregate variance. This type of aggregate-level leakage is similar in spirit to standard FL, where clients can observe changes in global model weights across rounds and infer coarse training dynamics.

---

> ### Author Response · Authors · 2026-04-26
> **Response to Reviewer Ri81 [3]**
>
> ### R.2 Contextualizing the theoretical bounds.
>
> **Response:** We revised the paper to position the theory as an analysis of the projected local training recursion. The bounds identify two projection-dependent quantities, and the new empirical estimates connect these quantities to the implemented SVD threshold.
>
> **What the bounds reveal.**
> Theorems 1–2 decompose FedProTIP's behavior into two projection-dependent quantities: $\rho_t$ (the fraction of current-task gradient energy preserved after projection) and $\beta_\tau^{(t-1)}$ (the fraction of old-task gradient energy remaining in the admissible update space). In the resulting bounds, the leading projection-dependent term in the convergence guarantee scales with $1/\rho_t$, while the forgetting bound scales with the average interference coefficient $\bar{\beta}_T$. Therefore, the analysis identifies the quantities that control the stability–plasticity trade-off under FedProTIP's per-step local projection. This differs from post-aggregation projection where local updates are unconstrained before aggregation and interference can accumulate over $S$ local steps. We do not claim that the bounds are numerically tighter than prior work; rather, they provide an explanation for why local projection is more robust under client heterogeneity, consistent with the empirical results in Figure 5.
>
> **Practical consequence.**
> We now provide empirical measurements of $\hat{\rho}_t$ and $\hat{\beta}$ across five threshold values (Figure 2; see P.3 for details). Both decrease monotonically with $\epsilon_l$, showing that the SVD threshold provides an empirical control knob for the trade-off between stability and plasticity identified by the bounds. This also helps explain why accuracy is relatively insensitive to $\epsilon_l$ (Table 11): lower $\hat{\rho}_t$ suggests slower convergence, while lower $\hat{\beta}$ suggests reduced forgetting, and these effects partially offset one another. The remarks in Section 5 have been revised to include this interpretation and reference the empirical analysis.

---

> > ### Author Response · Authors · 2026-04-26
> > **Response to Reviewer Ri81 [4]**
> >
> > ### R.3 FOT accuracy discrepancy.
> >
> > **Response:** The discrepancy comes from the inference protocol. The 71.9% reported in the FOT paper is under *task-aware* inference, where the ground-truth task ID routes predictions through the correct task-specific head. In contrast, Table 1 in our paper reports *task-agnostic* accuracy, where task IDs are unavailable at test time and methods must operate without oracle task routing. Under task-aware inference, our implementation of FOT achieves 82.59% (Table 16), which is consistent with the published FOT result after accounting for differences in backbone and hyperparameters. Table 1's caption now explicitly states that all numbers are task-agnostic, and Section 6.1 directs readers to Figure 3(b) for the task-aware comparison. The gap between FOT's task-aware and task-agnostic accuracy in our implementation (82.59% $\to$ 46.86%) reflects the absence of a task-identification mechanism, which is precisely the gap that FedProTIP's TIP module is designed to address.
> >
> > ---
> >
> > ### R.4 TIP when input distribution does not change.
> >
> > **Response:** TIP operates on *last-layer activations* $a_L$, not raw inputs. The relevant structural property is that task subspaces are constructed to be approximately orthogonal: the bases $U_L^{(t)}$ are extracted from activations projected against prior subspaces $\Phi^{(1:t-1)}$ (Eq. 6). TIP additionally requires that a test sample's activation have distinguishable energy across these task subspaces, so that the relevance scores $\omega^{(\tau,t)}$ differ across candidate tasks. This condition is naturally satisfied under the standard CL setting where tasks induce distinct data distributions $D^{(t)}$ (Section 3.1). Our experiments also probe settings where this assumption is partially relaxed:
> >
> > - *Class-overlapping tasks* (Table 15, Appendix D.6): with 1-class overlap, TIP achieves 83.90% versus 84.02% oracle; with 2-class overlap, TIP improves accuracy from 28.48% for the shared-head baseline to 76.97%.
> > - *Domain-incremental learning* (Table 1, DomainNet): all domains share the same 345-class label space, yet TIP achieves the lowest forgetting across all heterogeneity levels.
> >
> > The reviewer's adversarial construction, where tasks have identical $p(x)$ but different labeling semantics, is a limiting case in which the feature extractor may not develop distinguishable task-specific representations. In that case, relevance scores can become nearly uniform across task subspaces, and TIP is not expected to reliably identify the task. We added a paragraph in Section 4.4 clarifying this assumption and explicitly acknowledging this limitation.
> >
> > ---
> >
> > ### R.5 Complexity discussion and PEFT-based FCL.
> >
> > **Response:** We have (i) restructured Section 6.5 to separate asymptotic method overhead from empirical wall-time and GPU-memory measurements, adding Table 4 to compare per-layer computation, communication, and memory costs in big-$O$ notation, and (ii) added a paragraph in Appendix F.1 discussing parameter-efficient CL/FL/FCL methods such as InfLoRA, FlexLoRA, and PILoRA. FedProTIP operates in the full fine-tuning regime, while these methods are complementary approaches that are better suited to settings where a large pretrained backbone is frozen or adapted through a small number of trainable parameters.
> >
> > ---
> >
> > ### R.6 DomainNet: FedProTIP(-t) wins ACC but FedProTIP(+t) wins FT.
> >
> > **Response:** In domain-incremental learning, all tasks share the same label space. FedProTIP(-t) uses a shared classifier and can benefit from cross-domain transfer, which improves accuracy. FedProTIP(+t) instead uses separate task-specific classifiers routed by TIP; this isolates per-domain representations and reduces forgetting, but occasional routing errors can lower accuracy. We added a remark in Section 6.2 explaining this trade-off and clarifying that TIP is most beneficial in class-incremental settings, where task-specific heads are necessary.

---

### Review · Reviewer_KmvL · 2026-04-14

**Summary Of Contributions:**

The paper proposes FedProTIP, a replay-free federated continual learning method that projects gradients during local client training and uses aggregated activation subspaces for task-identity prediction at test time. The main empirical claim is that local projection is more robust than post-aggregation projection, especially under heterogeneity, and the reported experiments on CIFAR100, ImageNet-R, and DomainNet are strong. My main concern is that the mathematical theory is much weaker than the method claims: it supports the optimization intuition, but does not really justify the full activation-subspace and task-prediction pipeline.

Strengths
* The paper proposes a clean and well-motivated idea: projecting gradients during local client optimization rather than only after server aggregation, which is particularly sensible in federated continual learning under heterogeneous data.
* The method addresses the practically important setting of task-agnostic inference without relying on replay buffers or generative models.
* The empirical evaluation is solid, with strong results across multiple datasets, heterogeneity levels, and baseline methods.


Weaknesses
* The theoretical analysis is much less convincing than the empirical part, because it does not really prove correctness of the full method and instead relies on abstract quantities that are not tightly connected to the implemented algorithm.
* There is a mathematical gap between the activation-based subspace construction used in the method and the gradient-based quantities appearing in the theory.
*The task-identity prediction component is somewhat heuristic, and the paper does not fully explain why it works so well, especially in settings where it yields very large gains.

**Audience:**

Yes

**Audience Explanation:**

the paper addresses replay-free, task-agnostic federated continual learning, which is a relevant and active topic that should interest readers working on continual learning, federated learning, and privacy-constrained distributed optimization.

**Claims And Evidence:**

Yes

**Claims Explanation:**

the empirical results are strong and clearly presented, but the theoretical evidence is less convincing and does not fully justify the paper’s stronger mathematical claims.

**Requested Changes:**

Requested changes

* Please report the task-identity prediction accuracy explicitly in Section 6.1, since the large performance gap between FedProTIP and FedProTIP (-t) in Fig. 2 and Table 1 appears to come mainly from the TIP module.

* Please soften the theory claims in the Abstract, Introduction, and Section 5. In particular, Theorems 1--2 and Corollary 1 are written in terms of abstract coefficients \(\rho_t\) and \(\beta_{\tau}^{(t-1)}\), so they do not directly justify the activation-based subspace construction used in Sections 4.2--4.4.

* Please add a clearer discussion of the DomainNet results in Table 1, where the full method underperforms FedProTIP (-t) in accuracy. This is important because TIP is presented as a main contribution, so the paper should explain more clearly when it helps and when it may hurt.

Mathematical concerns / requested clarification

* Please clarify the gap between the algorithm and the theory: the method builds subspaces from activation SVDs in Sections 4.2--4.3, while Section 5 analyzes projected gradients through \(\rho_t\) and \(\beta_{\tau}^{(t-1)}\). The paper should state more explicitly that the current theory is only indirect support for the method, not a full mathematical justification.

* Please revise the discussion after Theorem 2 and Corollary 1, especially the claim that the theory explains the role of the projection threshold \(\epsilon_l\). As written, this connection is not established rigorously.

* Please clarify the rank-selection rule in Section 4.2, since the current singular-value thresholding criterion is unusual and should be stated more precisely for reproducibility.

---

> ### Author Response · Authors · 2026-04-26
> **Response to Reviewer KmvL**
>
> ## Summary of Changes
>
> - **TIP accuracy (K.1):** We now report per-task TIP routing accuracy in Figure 4(a) and Table 14, with direct reference in Section 6.1.
> - **Theory claims (K.2):** We revised the Remarks in Section 5 to present the connection between $\epsilon_l$ and the theoretical quantities as an empirical diagnostic supported by Figure 2.
> - **DomainNet trade-off (K.3):** We added discussion in Section 6.2 explaining that TIP is most beneficial in class-incremental settings; in domain-incremental settings, a shared head can yield higher accuracy while TIP achieves lower forgetting.
> - **Rank-selection rule (K.4):** We re-wrote Eq. (8) in Section 4.2, clarifying that $\epsilon_l \in (0,1]$ is a relative energy threshold.
>
> Several points overlap with those raised by Reviewers Ri81 and P8ee; where appropriate, we refer to the corresponding responses.
>
> ---
>
> ### K.1 Report TIP accuracy explicitly in Section 6.1.
>
> **Response:** Table 14 (Appendix D) now reports per-task TIP routing accuracy across all benchmarks. On 10-split CIFAR100, TIP achieves $\geq 0.978$ across all tasks; on 10-split ImageNet-R, $\geq 0.87$; and on 6-split DomainNet, $\geq 0.67$. Section 6.1 now references this table and Figure 4(a) directly, helping readers assess how task routing quality contributes to the performance gap between FedProTIP and FedProTIP(-t). See P.1 for further discussion.
>
> ---
>
> ### K.2 Soften theory claims and clarify the gap between activation-based subspace construction (Sections 4.2–4.3) and gradient-based theory (Section 5).
>
> **Response:** We address these two points jointly because they concern how the activation-based construction relates to the gradient-space quantities $\rho_t$ and $\beta_\tau^{(t-1)}$ in the bounds.
>
> The activation-based projector and the gradient-space quantities are connected through the layer-wise gradient structure. For a linear layer, the gradient decomposes as
> $\nabla_{W_l}\mathcal{L} = a_{l-1} \delta_l^\top$, where
> $a_{l-1}$ is the input activation and $\delta_l$ is the backpropagated error signal. Thus, the projection operator
> $P^{(t-1)} = I - \Phi^{(1:t-1)}(\Phi^{(1:t-1)})^\top$ acts on the activation factor,
> $P^{(t-1)}\nabla_{W_l}\mathcal{L} = (P^{(t-1)}a_{l-1}) \delta_l^\top$.
> The quantities $\rho_t$ and $\beta_\tau^{(t-1)}$ therefore measure how the activation-based projector interacts with the gradient field, rather than being defined in an unrelated space. We added a clarifying sentence in Section 5 to make this connection explicit.
>
> We agree, however, that the original manuscript overstated the connection between the threshold $\epsilon_l$ and these quantities. The revised remarks in Section 5 now provide empirical measurements of $\hat{\rho}_t$ and $\hat{\beta}$ across five threshold values (Figure 2), showing that both decrease monotonically with $\epsilon_l$. A larger protected subspace removes more activation directions from the gradient, reducing both $\hat{\rho}_t$ and $\hat{\beta}$ simultaneously. Because these opposing effects (slower convergence but less forgetting) partially offset one another, end-task accuracy remains largely insensitive to $\epsilon_l$, consistent with the ablation results (Figure 4(b)). We now present this relationship as an empirical diagnostic rather than a theorem-derived consequence.
>
> ---
>
> ### K.3 Discuss DomainNet results where FedProTIP underperforms FedProTIP(-t) in accuracy.
>
> **Response:** In domain-incremental learning, all tasks share the same label space. FedProTIP(-t) uses a shared classifier that can benefit from cross-domain knowledge transfer, yielding higher accuracy. FedProTIP(+t) uses separate task-specific classifiers routed by TIP, which isolates per-domain representations and achieves lower forgetting, but occasional routing errors can reduce accuracy. We added a remark in Section 6.2 explaining this trade-off and clarifying that TIP is most beneficial in class-incremental settings, where task-specific heads are necessary, whereas a shared head can be preferable in domain-incremental settings (Table 1, Figure 4(a)). See also R.6.
>
> ---
>
> ### K.4 Clarify the rank-selection rule in Section 4.2.
>
> **Response:** The original text omitted the denominator, making it ambiguous whether $\epsilon_l$ refers to an absolute or relative threshold. Eq. (8) in Section 4.2 has been rewritten to include the explicit ratio,
> $$r_l = \min \\{ r \in \mathbb{N}: \sum_{i=1}^{r} \sigma_{l,i} \big/ \sum_{i=1}^{\min(d_l, m_s)} \sigma_{l,i} \geq \epsilon_l \\}$$
> where $\sigma_{l,i}$ are the singular values of the projected activation matrix $\tilde{a}_l^{(t)}$ in descending order and $\epsilon_l \in (0,1]$ is a layer-specific relative energy threshold. This follows the relative-energy selection rule commonly used in GPM-style subspace methods.

---

### Author Response · Authors · 2026-06-20
**Camera-Ready Submission**

Dear Action Editor and Reviewers,

We sincerely thank the Action Editor for accepting our manuscript and for the encouraging final summary.

We are also deeply grateful to the reviewers for their constructive feedback throughout the review process. Your insights regarding the prior-work comparison with FOT, the precision of our privacy claims, the task-identity prediction (TIP) evaluation, and the contextualization of our theoretical analysis were invaluable. We believe that addressing these points has substantially strengthened the paper and clarified our contributions to the federated continual learning community.

We have uploaded the camera-ready version of the manuscript. Thank you once again for your time, effort, and guidance in improving our work.

Best regards,

The Authors

---

### Decision · Action_Editor_Ks5o · 2026-06-05

**Recommendation:** Accept as is

**Additional Comments:**

The paper has improved substantially after revision. The reviewers' main concerns have been addressed to a satisfactory degree: the prior-work comparison with FOT has been corrected, the privacy-related claims have been made more precise, a secure-aggregation-compatible sketching variant has been added, the TIP mechanism has been evaluated more explicitly, and the theory has been repositioned as providing insight into the stability-plasticity trade-off rather than a complete mathematical justification of the full pipeline.

The remaining limitations, including the heuristic nature of the TIP mechanism, the indirect connection between the theoretical quantities and the practical implementation, and the absence of formal privacy guarantees, are now sufficiently acknowledged and do not prevent acceptance under TMLR's criteria of soundness and interest to the audience. I therefore recommend acceptance.

**Audience:**

Yes

**Audience Explanation:**

The paper addresses an important and timely problem at the intersection of federated learning and continual learning: how to mitigate catastrophic forgetting in distributed sequential learning without relying on replay buffers, generative replay, or oracle task identities at inference time. This setting is relevant to privacy-constrained and resource-constrained deployment scenarios, where clients encounter evolving data distributions and cannot revisit past data.

The proposed combination of local gradient projection, compact subspace communication, and task-agnostic inference is likely to be of interest to researchers working on continual learning, federated learning, privacy-aware distributed optimization, and task-agnostic model adaptation. The empirical finding that local projection can be more robust than post-aggregation projection under client heterogeneity, together with the analysis of task-identity prediction and efficiency, provides useful evidence for the TMLR audience. Although the individual components are based on known techniques, their integration into a task-agnostic federated continual learning framework is a meaningful and practically motivated contribution.

**Claims And Evidence:**

Yes

**Claims Explanation:**

The reviewers generally agree that the revised submission provides sufficient evidence to support its main claims. The paper proposes FedProTIP, a replay-free federated continual learning framework that combines local gradient projection with task-identity prediction based on learned subspaces. The empirical evaluation covers multiple benchmarks, including CIFAR-100, ImageNet-R, and DomainNet, and compares against relevant federated continual learning baselines under different settings. The ablation studies further clarify the contributions of local projection, task-identity prediction, client heterogeneity, and efficiency.

The initial concerns mainly focused on the precision of the privacy discussion, the relationship between the theory and the implemented activation-subspace method, the task-identity prediction mechanism, the comparison with FOT, and the complexity analysis. In the revision, the authors corrected the description of FOT, added a secure-aggregation-compatible sketching variant, reported standalone TIP accuracy, clarified the task-agnostic evaluation protocol, discussed the DomainNet trade-off, separated asymptotic complexity from empirical measurements, and softened the theoretical claims. While the theory remains an indirect explanation rather than a full end-to-end guarantee for the complete pipeline, the revised paper is sufficiently clear about this limitation. Overall, the main claims are supported by accurate and convincing empirical evidence, with appropriate qualifications.